# Evolution of Cooperation in Public Goods Games with Stochastic Opting-Out

**Alexander G. Ginsberg [1,†] and Feng Fu [2,3,*]** 

1   Department of Mathematics, Michigan State University, East Lansing, MI 48824, USA; ginsbera@umich.edu
2   Department of Mathematics, Dartmouth College, Hanover, NH 03755, USA
3   Department of Biomedical Data Science, Geisel School of Medicine at Dartmouth, Lebanon, NH 03756, USA
*   Correspondence: fufeng@gmail.com; Tel.: +1-603-646-2293
†   Current address: Department of Mathematics, University of Michigan, Ann Arbor, MI 48109, USA

**Abstract:** We study the evolution of cooperation in group interactions where players are randomly drawn from well-mixed populations of finite size to participate in a public goods game. However, due to the possibility of unforeseen circumstances, each player has a fixed probability of being unable to participate in the game, unlike previous models which assume voluntary participation. We first study how prescribed stochastic opting-out affects cooperation in finite populations, and then generalize for the limiting case of large populations. Because we use a pairwise comparison updating rule, our results apply to both genetic and behavioral evolution mechanisms. Moreover, in the model, cooperation is favored by natural selection over both neutral drift and defection if the return on investment exceeds a threshold value depending on the population size, the game size, and a player's probability of opting-out. Our analysis further shows that, due to the stochastic nature of the opting-out in finite populations, the threshold of return on investment needed for natural selection to favor cooperation is actually greater than the one corresponding to compulsory games with the equal expected game size. We also use adaptive dynamics to study the co-evolution of cooperation and opting-out behavior. Indeed, given rare mutations minutely different from the resident population, an analysis based on adaptive dynamics suggests that over time the population will tend towards complete defection and non-participation, and subsequently cooperators abstaining from the public goods game will stand a chance to emerge by neutral drift, thereby paving the way for the rise of participating cooperators. Nevertheless, increasing the probability of non-participation decreases the rate at which the population tends towards defection when participating. Our work sheds light on understanding how stochastic opting-out emerges in the first place and on its role in the evolution of cooperation.

**Keywords:** adaptive dynamics; finite populations; social dilemmas; evolutionary dynamics; mathematical biology

## 1. Introduction

Cooperation is everywhere [1–4]. Bacteria cooperate. For example, bacteria cooperate in biofilm production, where bacteria go so far as to use quorum sensing to determine when there are enough cooperators that contributing to the biofilm is worthwhile [5]. Ants cooperate, building vast anthills where members of a colony live together [6]. Birds cooperate, sounding an alarm when predators are nearby [7]. Additionally, several felids and canids cooperate, working together to catch prey [8]. Moreover, humans cooperate [9]. Indeed, whenever we contribute to a joint hunting effort [10], bring food to a potluck, or work together to combat climate change [11], we are cooperating. Why, though, do we see cooperation in all walks of life? How does cooperation evolve?

Researchers have dedicated significant effort in the past decades towards studying the evolution of cooperation. See Traulsen and Nowak [3], Antal et al. [12], Boyd et al. [13], Broom and Rychtár [14], Hauert et al. [15], Hauert et al. [16], Javarone [17], Nowak [18], Priklopil et al. [19], Santos et al. [20] as examples.

In particular, one common type of social interaction in which cooperation frequently arises and which has recently attracted attention from researchers is the public goods game (in abbreviation as PGG thereafter). See [11,13,15,16,21–24]. In a PGG, cooperators contribute to a common pool which all participants of the game then share equally. In fact, in all of the instances of cooperation mentioned in the preceding paragraph, organisms contribute to a public good. In the case of bacteria, the public good is biofilm production. For ants, the good is the anthill. For birds, the good is the knowledge that a predator is nearby and hence that they should be careful. For felids and canids, the good is the catch of the hunt. Lastly, for the party-goers, the good is the food at the potluck.

However, whenever cooperators contribute to a common pool, there are free-riders, who benefit from the common pool without contributing. Game theorists frequently refer to such free-riders as *defectors*. These defectors cause the participants of the game to receive a smaller share of the common pool—a smaller payoff—than the social optimum where every player cooperates. In fact, assuming players can only cooperate or defect, a defector earns a larger payoff regardless of the number of cooperators because the defector does not have to contribute to the common pool, making defection the *dominant strategy* (or more generally individually optimal strategy). Game theorists refer to the situation in which the dominant strategy is individually optimal yet not socially optimal as a *social dilemma* [25]. Consequentially, if each player were rational, each player would choose to defect, regardless of the strategies of other players. As a result, each player would receive zero payoff worse than that if all have cooperated otherwise. This outcome is often called as *the tragedy of the commons* [1].

In reality, even though in any particular PGG defectors outperform cooperators in individual games, it may be the case that cooperators may actually outperform defectors, when averaging over all possible games. Such situation is an example of the Simpson's paradox [16]. Additionally, there are many ways in which a tweak to the PGG may promote cooperation [26–31]. For instance, spatial selection [18,32] or more generally population structure [12,14,20,33–40], punishment of defectors [13], signaling [23,41], and optional participation [15,16], and combinations of the latter two mechanisms [24,42] have been used to promote cooperation. However, in the literature, a small but realistic tweak to the PGG has yet to be addressed. Specifically, even if there is no punishment of defectors or if players cannot opt-out, due to unforeseen circumstances, at times players simply cannot participate in the PGG. For instance, an individual traveling to a hunting party may come across a flooded road and be forced to turn back. Or, on a whim, an individual may decide to engage in some activity other than the game. Further, a player could be late to the game, or fall ill, missing out on the opportunity to participate entirely. Alternatively, the game could draw all individuals in a given area, in which case we could define game size as the number of players that frequent that area. In this case, we expect players that frequent the area to randomly choose not to pass through the area when the PGG occurs. As a result, players participate in the PGG stochastically, unable to participate independently of whether or not the player plans to cooperate or defect.

We investigate evolutionary dynamics of such PGGs with stochastic non-participation. We add a fully analyzed stochastic model to the literature, thus improving the understanding of the evolution of cooperation. Moreover, our model demonstrates that the evolution of cooperation can be promoted by the stochasticity in participation. We conclude with an analysis of adaptive dynamics for simplified two-person PGGs in finite populations, where we add rare and minute mutations to our basic model. We identify the condition for non-participation to be favored in the coevolutionary dynamics of cooperation and opting-out behavior. We also find that increasing the probability of non-participation temporarily slows the rate at which the population tends to defection when participating given rare mutations only minutely different from the resident population.

## 2. Model and Methods

We consider a well-mixed finite population of *n* agents, human or not, and suppose that frequently $N \leq n$ randomly selected agents receive the opportunity to participate in a PGG (Figure 1). In the PGG, some participants cooperate, either by conscious choice or by their genetics, investing one unit into a common pool, as in [15,16]. Every unit contributed by each cooperator is multiplied by some factor *r* (return on investment or enhancement factor), $1 < r < N$, and thus for each unit contributed by a cooperator, the common pool increases by *r* units. At the end of the game, each PGG participant obtains an equal share of the common pool. However, the participants who do not contribute (namely, defectors) also receive a share from the common pool by free-riding. To simplify the model, we assume that participants determine their strategies before the PGG has begun (that is, not dependent on group composition), as in [15,16].

As stated, the preceding model of PGG interaction leads to domination by defectors for all games where the return on investment *r* is smaller than the group size *N* and the game is thus a social dilemma. To promote cooperation, we assume that due to unforeseen circumstances each selected agent has a fixed probability *α* of being unable to participate in the PGG, instead obtaining a fixed payoff $\sigma > 0$ (often called the loner's payoff [16]). Indeed, in many games there is no good reason that all *N* selected agents with the opportunity and desire to participate in the game should be guaranteed to participate. Hence, by representing the probability that any selected agent will be unable to play due to such unforeseen circumstances, introducing *α* makes our model more realistic (Figure 1).

Additionally, our model needs an "update mechanism" by which the population may change its composition of agents that are cooperating or defecting. We use the pairwise comparison rule as in previous studies [23,24,43], where two agents are randomly selected, and one agent, the focal "updating agent", will update his or her strategy by comparing his or her payoff to the other agent, the "compared agent". We may think of this "update" either as the conscious choice of the updating agent to change strategy (social imitation) or as the death of the updating agent and subsequent replacement by an offspring of the compared agent (death-birth process) [25].

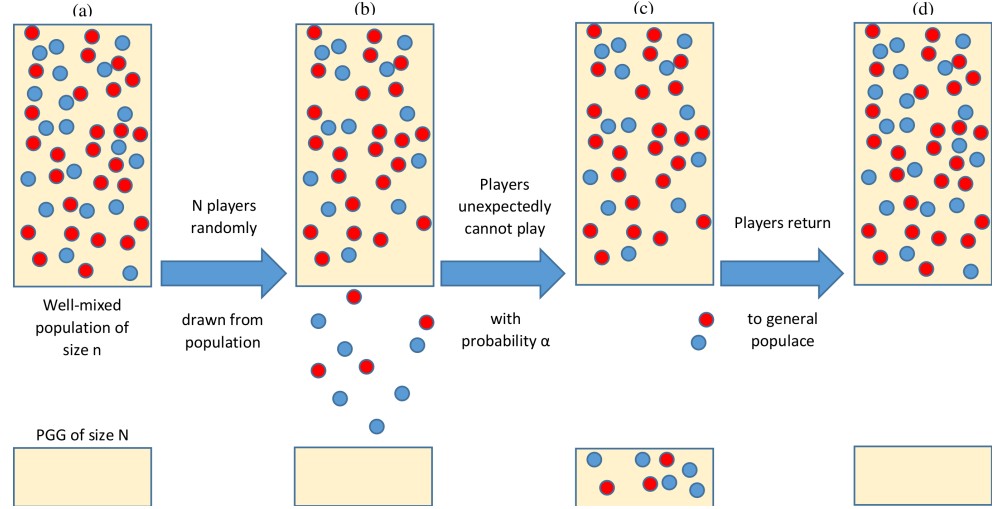

**Figure 1.** Model schematic of stochastic opting-out. Cooperators (blue) and defectors (red) are represented by dots. A fixed number of agents are randomly drawn from the population to participate in a PGG, represented by the small tan rectangular area. While most selected agents are able to make it to the game, some are not. Selected agents then return to the general populace, where no game is occurring. The fitness of agents is determined by the average payoffs they obtain from PGG interactions (cooperators vs. defectors) as well as from non-participation. Natural selection drives the co-evolutionary dynamics of opting-out behavior (the probability of non-participation, *α*), and cooperation (the probability to cooperate in the PGG, *β*).

Then, let $p_{i\leftarrow j}$ be the probability that the updating agent, $i$, adopts the strategy of the compared agent, $j$. Specifically, we let the probability $p_{i\leftarrow j}$ of adopting strategies be given by the Fermi function, as in [23,24,43]:

$$p_{i\leftarrow j} = \frac{1}{1 + \exp[-\gamma(\pi_j - \pi_i)])}, \tag{1}$$

where $\pi_j$ represents the expected payoff of the compared agent $j$, $\pi_i$ represents the expected payoff of the updating agent $i$, and $\gamma \geq 0$ represents the selection pressure and corresponds to the inverse temperature in statistics physics [43].

When it comes to adaptive dynamics in finite populations, for simplicity, we assume the PGG size is two. Furthermore, applying adaptive dynamics to our model as done in Imhof and Nowak [44], we assume that a single mutant who plays a different strategy but sufficiently close to the resident population attempts to invade the population. Specifically, we suppose that every agent uses a strategy in the prescribed strategy space $(\alpha, \beta)$, where $\alpha$ is the probability that due to unforeseen circumstances the selected agent cannot participate, and $\beta$ is the probability that the agent cooperates if they participate. We let the original population be composed solely of agents with strategy $(\alpha, \beta)$, and we suppose that the population is invaded by a single agent with strategy $(\alpha', \beta')$. Then, we let $\alpha' \to \alpha$, and $\beta' \to \beta$. As in Imhof and Nowak [44], we also assume rare and minute mutations. That is, we assume sufficient time passes between mutations that either fixation, or extinction, of the mutant type occurs, and that the invading mutant population plays strategy only minutely different from the resident population.

## 3. Results

### 3.1. Pairwise Invasion Dynamics in Finite Populations

To proceed with the analysis of the stochastic model, let us first calculate the expected payoffs for cooperators and defectors, $\pi_c$ and $\pi_d$, respectively. To calculate $\pi_d$, we use the method presented by Hauert et al. [15]. First, we observe that in a game with $S$ players, defectors receive a benefit $rn_c/S$, where $n_c$ is the number of cooperators in the game, if $S > 1$. However, if $S = 1$, that player must be a loner (because of excluding self-interactions), and will receive the loner's payoff, $\sigma$. Then, noting that any player does not play with probability $\alpha$ and plays with probability $1 - \alpha$, and letting $x_c$ be the proportion of cooperators in the population, we get

$$\pi_d = \alpha\sigma + (1 - \alpha)[rx_c[1 - (1 - \alpha^N)/[N(1 - \alpha)]] + \alpha^{N-1}\sigma]. \tag{2}$$

We defer the details of calculating $\pi_d$ to Appendix A. Employing a similar method (see details in Appendix B), we obtain

$$\pi_c = \pi_d - r/(n-1)[1 - \alpha - (1 - \alpha^N)/N] + (1 - \alpha)[-1 + (1 - r)\alpha^{N-1} + (r/N)(1 - \alpha^N)/(1 - \alpha)]. \tag{3}$$

Hence,

$$\begin{aligned} \pi_c - \pi_d = &- r/(n-1)[1 - \alpha - (1 - \alpha^N)/N] \\ &+ (1 - \alpha)[-1 + (1 - r)\alpha^{N-1} + (r/N)(1 - \alpha^N)/(1 - \alpha)], \end{aligned} \tag{4}$$

which is a constant with respect to the proportions of cooperators and defectors in the population.

Then, substituting $\pi_c - \pi_d$ into Equation (1), we obtain the probability that a cooperator becomes a defector or that a defector replaces a cooperator, given that a cooperator is selected for updating and a defector is selected for comparison, is

$$\begin{aligned} p_{cd} = &(1 + \exp[\gamma(-r/(n-1)[1 - (1 - \alpha^N)/[N(1 - \alpha)]] + \\ &(1 - \alpha)[-1 + (1 - r)\alpha^{N-1} + (r/N)(1 - \alpha^N)/(1 - \alpha)]])^{-1}. \end{aligned} \tag{5}$$

Thus the probability that the number of cooperators decreases by one in one iteration of the pairwise comparison updating process given $i$ cooperators is

$$T_i^- = \frac{i}{n}\frac{n-i}{n}p_{cd}. \tag{6}$$

Likewise, the probability that a defector becomes a cooperator or that a cooperator replaces a defector given that the defector is selected for updating and the cooperator is selected for comparison is

$$\begin{aligned}p_{dc} =&(1+\exp[-\gamma(-r/(n-1)(1-\alpha)[1-(1-\alpha^N)/[N(1-\alpha)]]+\\ &(1-\alpha)[-1+(1-r)\alpha^{N-1}+(r/N)(1-\alpha^N)/(1-\alpha)]])^{-1},\end{aligned} \tag{7}$$

which is also a constant. Hence, the probability that the number of cooperators increases by one in one iteration of the pairwise comparison updating process is

$$T_i^+ = \frac{n-i}{n}\frac{i}{n}p_{dc}. \tag{8}$$

Of course, though, if the number of cooperators, $i$, is 0 or $n$, the probabilities that a cooperator will change to a defector and that a defector will change to a cooperator are both zero, and the number of cooperators remains at 0 or $n$. That is, $i = 0$ and $i = n$ are absorbing states in the model (in the absence of mutations in strategy updating).

Moreover, now knowing $p_{cd}$ and $p_{dc}$, and noting that $p_{cd} + p_{dc} = 1$, we can calculate the transition matrix $P$ for the Markov chain in which the pairwise comparison updating process is iterated repeatedly. However, as the transition matrix itself is not vital for our analysis, we defer discussion of the transition matrix to Appendix C. On the other hand, the fixation probability of cooperation, that is, the probability that given $i$ cooperators in a population of $n - i$ defectors that every individual will become a cooperator, *is* vital. Following the procedure outlined by [25], we show that the fixation probability of cooperation given $i \geq 1$ cooperators, $x_i$, is

$$x_i = (1+\Sigma_{j=1}^{i-1}\Pi_{k=1}^j p_{cd}/p_{dc})/(1+\Sigma_{j=1}^{n-1}\Pi_{k=1}^j p_{cd}/p_{dc}), \tag{9}$$

where $i = 1$ implies the numerator is 1 and $p_{cd}$ and $p_{dc}$ should be indexed by the number of cooperators $k$. However, since $p_{cd}$ and $p_{dc}$ are constants with respect to the number of cooperators $k$ as shown above, we have omitted the index $k$ in Equation (9) for notational simplicity. Further simplifying, denote $p_{cd}/p_{dc}$ by $G(\alpha, \gamma, N, n, r)$, and observe that

$$\begin{aligned}G(\alpha,\gamma,N,n,r) &= (1+\exp(-\gamma(\pi_c-\pi_d)))/(1+\exp(\gamma(\pi_c-\pi_d)))\\ &= \exp[-\gamma(\pi_c-\pi_d)].\end{aligned} \tag{10}$$

Since $G$ is constant over $i$, we may expand the numerator and denominator of $x_i$ as geometric series. So, if $G \neq 1$,

$$x_i = (1-G^i)/(1-G^n). \tag{11}$$

However, $G = 1$ means that $p_{cd} = p_{dc} = 1/2$, which implies neutral drift. We assume for now that $G \neq 1$. Then, observing that $p_{dc}/p_{cd} = G^{-1}$, the fixation probability of defection given $i$ defectors is simply $x_i$ with $G$ replaced by $G^{-1}$:

$$y_i = [G^{n-i}-G^n]/[1-G^n]. \tag{12}$$

Hence, the fixation probability given $i$ cooperators is

$$y_{n-i} = [G^i-G^n]/[1-G^n]. \tag{13}$$

Thus, the probability of fixation of cooperators or defectors given $i$ cooperators satisfies

$$x_i + y_{n-i} = 1. \tag{14}$$

In other words, the system always reaches an absorption state.

Furthermore, now knowing the probabilities of fixation of cooperation given $i$ cooperators, $x_i$, and of defection given $i$ defectors, $y_i$, we can determine which strategy is favored by natural selection. Moreover, as in Nowak [25], natural selection favors cooperation over defection if and only if $x_1 > y_1$ under pairwise invasion dynamics. Likewise, natural selection favors defection over cooperation if and only if $y_1 > x_1$ Nowak [25]. Additionally, natural selection favors cooperation over neutral drift if and only if $x_1 > 1/n$, where $1/n$ is the probability of fixation given neutral drift Nowak [25]. Likewise, natural selection favors defection over neutral drift if and only if $y_1 > 1/n$. In fact,

$$x_1 > 1/n \Leftrightarrow G < 1. \tag{15}$$

We defer the proof to Appendix D. Also, $G = 1$, implies neutral evolution, since $G = 1$ means $p_{cd} = p_{dc} = 1/2$. Since $G \neq 1$ implies either $p_{cd} > p_{dc}$ or vice-versa, there is neutral drift if and only if $G = 1$. Thus, $x_1 < 1/n$ if and only if $G > 1$.

Hence, natural selection favors cooperation over neutral drift if and only if $G < 1$, and disfavors cooperation if and only if $G > 1$. For $G = 1$ we have neutral evolution between cooperation and defection.

On the other hand, we can show that

$$G < 1 \Rightarrow y_1 < 1/n, \tag{16}$$

and

$$G > 1 \Rightarrow y_1 > 1/n. \tag{17}$$

We defer proofs of the two preceding assertions to Appendix D.

Additionally if $G = 1$, then there is neutral drift, as we demonstrated above, so $y_1 = 1/n$. Thus, if $G > 1$, $y_1 > 1/n > x_1$; if $G = 1$, then $y_1 = 1/n = x_1$; and otherwise, i.e., $0 < G < 1$, $y_1 < 1/n < x_1$.

Additionally, using Equation (10), it leads to

$$G > 1 \Leftrightarrow \pi_c - \pi_d < 0. \tag{18}$$

Likewise,

$$G < 1 \Leftrightarrow \pi_c - \pi_d > 0. \tag{19}$$

Also, $G = 1$ if and only if $\pi_c - \pi_d = 0$. Thus, there are three possibilities:

1.   Natural selection favors cooperation over defection $x_1 > 1/n > y_1$, if $\pi_c - \pi_d > 0$;
2.   Neutral evolution $x_1 = 1/n = y_1$, if $\pi_c - \pi_d = 0$;
3.   Natural selection favors defection over cooperation $x_1 < 1/n < y_1$, if $\pi_c - \pi_d < 0$.

Thus, the sign of $\pi_c - \pi_d$, as given in Equation (4), which is a function of the probability that a given player opts out $\alpha$, the PGG size $N$, the population size $n$, and the return on investment by cooperators, $r$, exclusively determines which strategies, cooperation or defection, natural selection favors more and whether or not natural selection favors each strategy replacing the other (Figure 2).

Notably, as shown in Figure 2, the graphs for $\pi_c - \pi_d < 0$ (Figure 2c,d), may be obtained from the graphs for $\pi_c - \pi_d > 0$ simply by relabeling cooperators as defectors and vice versa (Figure 2a,b). This is because reversing the sign of $\pi_c - \pi_d$ is equivalent to inverting $p_{cd}/p_{dc}$.

Based on these calculations of fixation probabilities as mentioned above, let us now identify the critical condition in terms of the threshold value of $r$, denoted by $R(\alpha)$, for given PGG size $N$,

above which cooperation will be favored by natural selection. It is easy to check that $r > R$ implies that $\pi_c - \pi_d > 0$, and $r < R$ implies that $\pi_c - \pi_d < 0$.

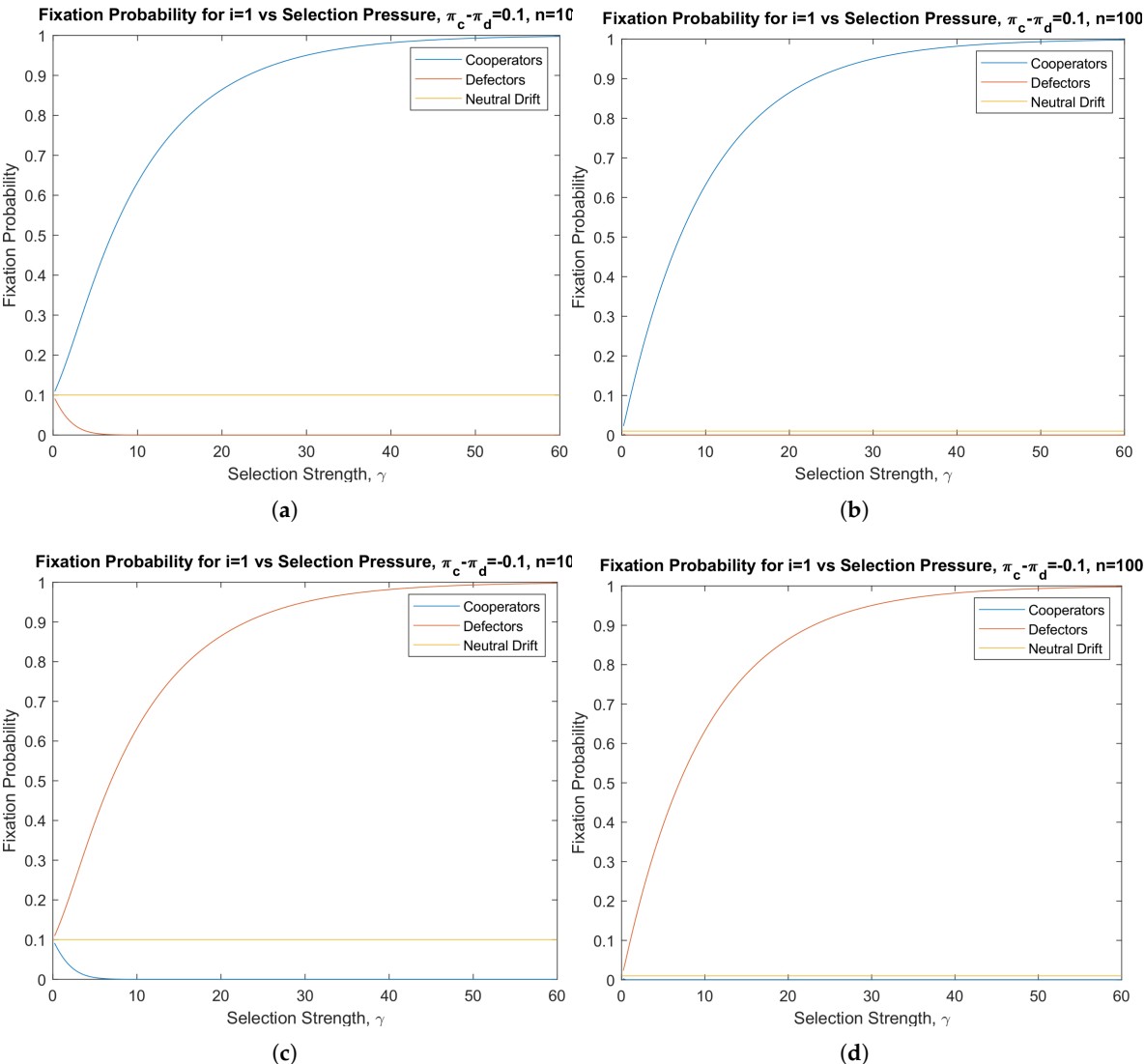

**Figure 2.** Pairwise invasion dynamics in finite populations. Shown are the graphs of fixation probabilities for $\pi_c - \pi_d > 0$, as in panels (**a**,**b**), and for $\pi_c - \pi_d < 0$, as in panels (**c**,**d**). If $\pi_c - \pi_d > 0$, the fixation probability starting with one cooperator $x_1$ is always larger $1/n$ (neutral drift) which is in turn always larger than that of one defector, $y_1$. On the other hand, if $\pi_c - \pi_d < 0$, then the situation is reversed, that is, $y_1 > 1/n > x_1$. We confirm that the specific values chosen for $\pi_c - \pi_d$ are admissible for the given values of population size $n$.

Indeed, recalling that $\pi_c - \pi_d = -r/(n-1)(1-\alpha)[1 - (1-\alpha^N)/[N(1-\alpha)]] + (1-\alpha)[-1 + (1-r)\alpha^{N-1} + (r/N)(1-\alpha^N)/(1-\alpha)]$ as given in Equation (4), it follows immediately that

$$\pi_c - \pi_d > 0 \Leftrightarrow \tag{20}$$

$$r > \frac{(1 - \alpha^{N-1})}{-\dfrac{1}{n-1}\left(1 - \dfrac{1-\alpha^N}{N[1-\alpha]}\right) + \dfrac{1-\alpha^N}{N(1-\alpha)} - \alpha^{N-1}} = R(\alpha), \tag{21}$$

provided of course that the denominator of the above expression is defined.

Further simplifying, we obtain

$$R(\alpha) = N \frac{1 - \alpha - \alpha^{N-1} + \alpha^N}{1 - \frac{N-1}{n-1} + \frac{N}{n-1}\alpha - N\alpha^{N-1} + \alpha^N(N - 1 - \frac{1}{n-1})}. \tag{22}$$

As shown in previous studies [25,39,42,43], population size has an impact on stochastic evolutionary dynamics in finite populations. We note that Equation (22) above explicitly shows how the conditions for cooperation to be favored depend on the population size $n$. We discuss the limit of large populations in Section 3.2 below.

The critical threshold $R(\alpha)$, as shown in Appendix E, is in fact defined necessarily on $[0, 1)$. Specifically, $R(\alpha)$ is undefined at $\alpha = 0$ if and only if $n = N$, in which case $R \to +\infty$ as $\alpha \to 0$, as shown in Lemma 3 of Appendix D. On the other hand, if $n > N$ (the PGG size $N$ less than the population size $n$),

$$R(0) = \frac{N(n-1)}{n-N}, \tag{23}$$

also as shown in Lemma 3 of Appendix D. Additionally, although $R(\alpha)$ is undefined at $\alpha = 1$ because there is essentially no game at $\alpha = 1$ as everyone is opting out, we can show that it has a left-handed limit at 1 as long as the population size $n > 2$:

$$\lim_{\alpha \to 1^-} R(\alpha) = \frac{2(n-1)}{n-2}, \tag{24}$$

In any case, $\pi_c - \pi_d > 0 \Leftrightarrow r > R(\alpha)$, wherever $R$ is defined. By analogous proofs, $\pi_c - \pi_d = 0$ if and only if $r = R(\alpha)$ and $\pi_c - \pi_d < 0$ if and only if $r < R(\alpha)$. Moreover, as proven in Appendix E, $R(\alpha)$ is strictly decreasing on $(0, 1)$ (Figure 3). Thus, for a given investment return factor $r$, there exists a threshold $\alpha_0$ satisfying $r = R(\alpha_0)$, such that increasing the likelihood of opting-out $\alpha > \alpha_0$ makes natural selection favor cooperation.

We note that this threshold $\alpha_0$ is analogous to the threshold on the proportion of individuals who choose to opt-out as suggested by Hauert et al. [15], which deals with an infinite rather than finite population and with planned, rather than unplanned stochastic, non-participation. This is an important insight from our present study about the impact of stochasticity in participation on the evolution of cooperation. Moreover, because this threshold $\alpha_0$ satisfies $r = R(\alpha_0)$, for increasing values of $\alpha$, the requirements on $r$ such that natural selection favors cooperation become less and less stringent. In other words, increasing the probability of non-participation facilitates cooperation (Figure 3).

*3.2. Approximations of the Critical Threshold $R(\alpha)$ for Natural Selection to Favor Cooperation*

In order to further gain an intuitive understanding of the critical threshold $R(\alpha)$, as given in Equation (22), for natural selection to favor cooperation, let us consider some limiting cases. To this end, we first suppose that the PGG size, $N$, is set as a fixed proportion of the population size, $n$. That is, $N = c(n - 1)$, where $c$ is a constant ratio.

We first suppose $n > N \gg 1$, as in shown Figure 3b. Then, as detailed in Appendix F.1, we find that

$$R(\alpha) \approx N(1 - \alpha)/(1 + c\alpha), \tag{25}$$

which is an approximation more manageable than the true $R(\alpha)$ in Equation (22). Next, we consider $n \gg N \gg 1$, as in shown in Figure 3d. Here we may let $c \to 0$ in Equation (25), obtaining

$$R(\alpha) \approx N(1 - \alpha). \tag{26}$$

Notably, we may lump these two limiting cases into the broader case in which $N \gg 1$ is required, and as proven in Appendix F.3, we have

$$R(\alpha) \to \frac{(n-1)N(1-\alpha)}{n - N(1-\alpha)} = R_{exp}(\alpha), \tag{27}$$

where we denote this approximation by $R_{exp}(\alpha)$.

Namely, $R_{exp}(\alpha)$ is the approximate threshold, which is required for natural selection to favor cooperation, in the limit of large PGG size $N$. Interestingly, this approximation (27) can be intuitively understood as the critical condition of $\pi_c > \pi_d$ in a population of $n$ agents playing the compulsory PGG with the fixed group size $N(1-\alpha)$, which is equal to the expected PGG size with stochastic opting-out. Hence, $R_{exp}(\alpha)$ can be obtained by using Equation (23) as $\alpha \to 0$ and then replacing $N$ by $N(1-\alpha)$.

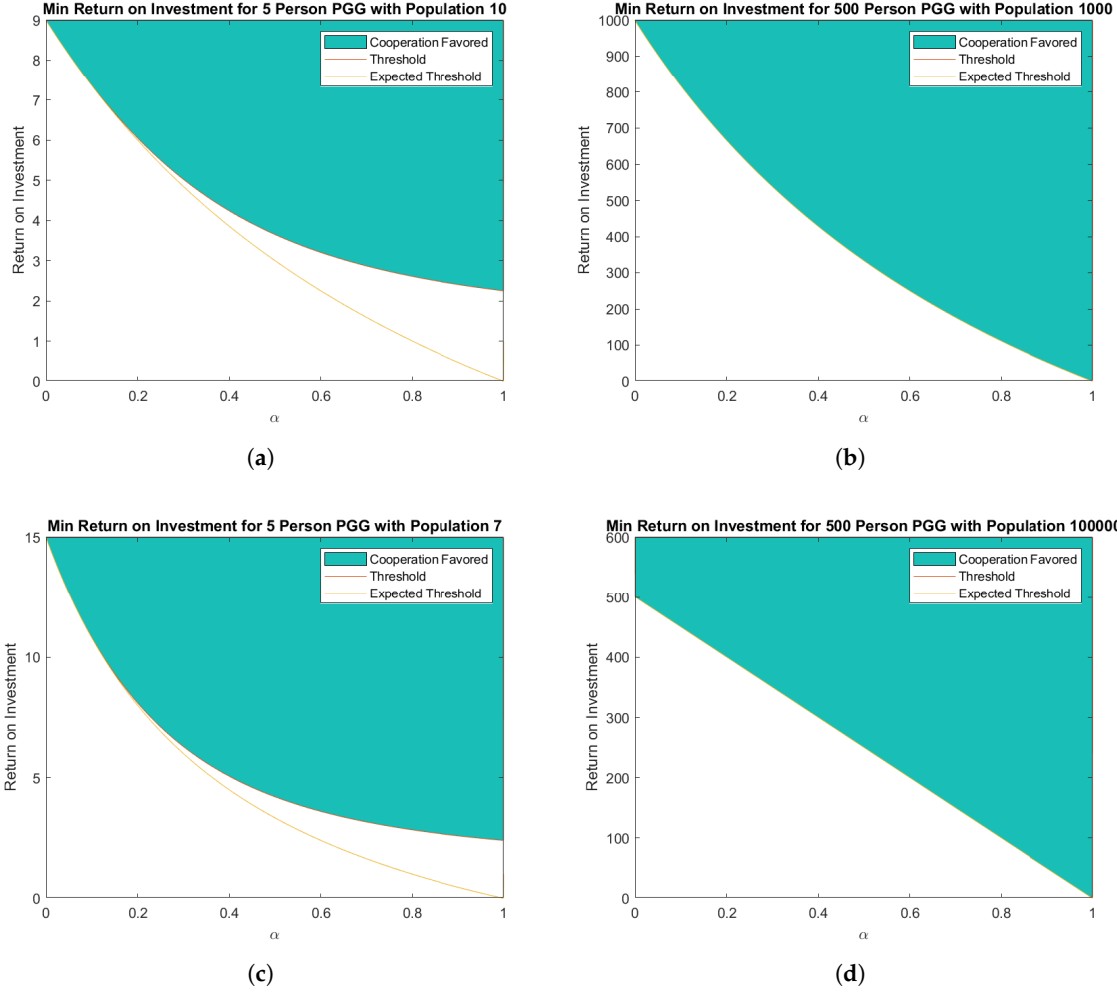

(a)

(b)

(c)

(d)

**Figure 3.** Critical threshold $R(\alpha)$ of the PGG investment return (i.e., enhancement factor) $r$ required for cooperation to be favored. The shaded areas represent combinations of the PGG enhancement factor $r$ and the probability of non-participation (i.e., opting-out) $\alpha$ which promote cooperation for game size $N = 5$ and $n = 10$ in (**a**), $N = 500$ and $n = 1000$ in (**b**), $N = 5$ and $n = 7$ in (**c**), and $N = 500$ and $n = 1,000,000$ in (**d**). The yellow line in each panel is the approximation $R_{exp}(\alpha)$ as given in Equation (27).

Lastly, we suppose $n \gg N > 1$ and let $c = N/(n-1) \to 0$. Then, using Equation (22), we get

$$R(\alpha) \approx N \frac{1 - \alpha - \alpha^{N-1} + \alpha^N}{1 - \alpha^{N-1}N + \alpha^N(N-1)} > N(1 - \alpha). \tag{28}$$

The inequality above is proven in Appendix D. Interestingly, as $\alpha \to 1$, by Equation (24), we know that $R$ tends to $2(n-1)/(n-2)$. Therefore, $\lim_{\alpha \to 1^-, n \to \infty} R(\alpha) \to 2$ (also see Appendix F.4).

*3.3. Adaptive Dynamics in Finite Populations*

Of particular interest is to study coevolutionary dynamics of cooperation (the probability to cooperate if participating in the PGG, $\beta$) and opting-out (the probability of non-participation, $\alpha$) in finite populations using the approach of adaptive dynamics in the continuous strategy space $(\alpha, \beta)$, represented by the unit square $[0,1] \times [0,1]$. To obtain closed-form results, here we consider the simplest possible case yet without loss of generality, that is, the two-person PGG ($N = 2$). In this case, the game in fact becomes an optional Prisoner's Dilemma (see Appendix G). (The more general PGGs, $N > 2$, can also be analyzed analogously by the method given here.)

Let us consider a population consisting of two types of players, the invaders with the mutant strategy $y$ and the resident population with the original strategy $x$, whom we call the defenders (also called the wild-type population), defined by their strategies $(\beta', \alpha')$ and $(\beta, \alpha)$, respectively. Unless otherwise noted, we maintain the notation used in Section 3.1, and we find the expected payoff for invaders is

$$\pi_y(i) = \frac{n-i}{n-1}(1-\alpha')(1-\alpha)(r\beta/2 + r\beta'/2 - \beta') + \frac{i-1}{n-1}(1-\alpha')^2\beta'(r-1) + \tag{29}$$
$$\sigma\alpha' + \sigma(1-\alpha')(\frac{n-i}{n-1}\alpha + \frac{i-1}{n-1}\alpha').$$

We defer the derivation of the expected payoff for invaders to Appendix G. Moreover, since the game is symmetric, the expected payoff for defenders may be determined simply by replacing the number of invaders, $i$, with the number of defenders, $n - i$, and by relabeling as appropriate. Specifically, the expected payoff for defenders is:

$$\pi_x(i) = \frac{i}{n-1}(1-\alpha)(1-\alpha')(r\beta'/2 + r\beta/2 - \beta) + \frac{n-i-1}{n-1}(1-\alpha)^2\beta(r-1) + \tag{30}$$
$$\sigma\alpha + \sigma(1-\alpha)(\frac{i}{n-1}\alpha' + \frac{n-i-1}{n-1}\alpha).$$

Continuing to use the pairwise comparison updating process, the probability that the number of invaders reduces by one, $T_i^-$, (where an invader is randomly chosen and adopts the strategy of a defender or is replaced by the offspring of a defender) is

$$T_i^- = p_{y \leftarrow x}(i) = \frac{i}{n}\frac{n-i}{n}\frac{1}{1 + \exp[-\gamma(\pi_x(i) - \pi_y(i))]}, \tag{31}$$

where $\gamma$ is the selection pressure, just as in Section 3.1. Similarly, the analogous probability that the number of invaders increases by one, $T_i^+$, is

$$T_i^+ = p_{x \leftarrow y}(i) = \frac{n-i}{n}\frac{i}{n}\frac{1}{1 + \exp[-\gamma(\pi_y(i) - \pi_x(i))]}. \tag{32}$$

Then, the fixation probability of an invader given $i$ invaders in a population of defenders is

$$x_i = (1 + \Sigma_{j=1}^{i-1}\Pi_{k=1}^{j} p_{y\leftarrow x}(k)/p_{x\leftarrow y}(k))/(1 + \Sigma_{j=1}^{n-1}\Pi_{k=1}^{j} p_{y\leftarrow x}(k)/p_{x\leftarrow y}(k)), \qquad (33)$$

where the backward-to-forward transition probability ratio is $p_{y\leftarrow x}(k)/p_{x\leftarrow y}(k) = \exp[\gamma(\pi_x(k) - \pi_y(k))]$.

To investigate the adaptive dynamics in finite populations [44], we consider

$$(\frac{d\alpha}{dt}, \frac{d\beta}{dt}) = \vec{f}(\alpha, \beta) = \lim_{(\alpha',\beta')\to(\alpha,\beta)} (\partial x_1/\partial\alpha', \partial x_1/\partial\beta'). \qquad (34)$$

The direction given by $\vec{f}$ for $(\alpha, \beta)$, plotting $\vec{f}$ as a vector field, is the direction in the strategy space which maximizes the fixation probability of *one* single invading mutant, $x_1$ (see Figure 4). Following the directions which maximize $x_1$ in the strategy space $(\alpha, \beta)$ starting at an initial $(\alpha_0, \beta_0)$, that is, following the streamlines of $\vec{f}$, indicates the most likely path in the strategy space that a population will take as mutants with similar strategies eventually fixate in the population, as suggested by Imhof and Nowak [44].

Substituting Equation (33) for $i = 1$, $x_1$, into Equation (34), and further simplifying, we obtain:

$$\frac{d\alpha}{dt} = \lim_{(\alpha',\beta')\to(\alpha,\beta)} \frac{\partial x_1}{\partial\alpha'} = (1-\alpha)\gamma(n-2)[\sigma - (r-1)\beta]/(2n), \qquad (35)$$

and

$$\frac{d\beta}{dt} = \lim_{(\alpha',\beta')\to(\alpha,\beta)} \frac{\partial x_1}{\partial\beta'} = (1-\alpha)^2\gamma(2 - 2n - 2r + nr)/(4n). \qquad (36)$$

We can see that increasing the likelihood of opting out, $\alpha$, slows down the overall rate of adaptation, which is given by the magnitude of the vector $\vec{f} = (\frac{d\alpha}{dt}, \frac{d\beta}{dt})$. This is because $\frac{d\alpha}{dt}$ is linearly dependent on $(1 - \alpha)$, as shown in Equation (35) and $\frac{d\beta}{dt}$ is quadratic in terms of $(1 - \alpha)$, as shown in Equation (36). Moreover, Equation (36) indicates that increasing the probability of stochastic opting-out, $\alpha$, can diminish the rate at which individuals in the population tend towards complete defection. We refer to Figure 4 for adaptive dynamics with various combinations of $r$ and $\sigma$ in a population of finite size $n$.

If the return on investment $r < (2n - 2)/(n - 2)$, for $\alpha < 1$, we have

$$\frac{d\beta}{dt} = \lim_{(\alpha',\beta')\to(\alpha,\beta)} \frac{\partial x_1}{\partial\alpha'} < 0, \qquad (37)$$

which means that the level of cooperation $\beta$ can be eroded gradually in the absence of complete non-participation ($\alpha < 1$), leading to complete defection in the long run.

Furthermore, if the payoff for non-participation $\sigma < r - 1$, there exists a critical threshold $\beta^* = \sigma/(r - 1) \in (0, 1)$ dividing the strategy space into two parts (as shown in Figure 4, panels b2 and c2): for $\beta > \beta^*$, we have $\frac{d\alpha}{dt} < 0$, which suggests that adaptive dynamics can favor mutant strategies with increasing the likelihood of participation (i.e., smaller values of $\alpha$); in contrast, for $\beta < \beta^*$, adaptive dynamics will favor opting-out as $\frac{d\alpha}{dt} > 0$.

Altogether, these results imply a cyclic population dynamics of cooperation, defection, and opting-out from the perspective of adaptive dynamics in finite populations. Complete opting-out strategies $(1, \beta)$ with high cooperativity $\beta > \beta^*$ are not evolutionarily stable and can be invaded by these strategies $(\alpha, \beta)$ with smaller likelihood of opting out and lower cooperativity. However, once cooperativity $\beta$ drops below $\beta^*$ in the population, strategies with increasing probability of non-participation will be favored. Going further, the population will either hit the edge $\beta = 0$ with zero cooperation first and move along this edge towards complete opting-out, that is, the corner $(1, 0)$ as $\frac{d\alpha}{dt} > 0$, or possibly the population will hit the edge $\alpha = 1$ first and remain there with zero participation yet having non-zero cooperativity $0 < \beta < \beta^*$. We note that $(1, 0)$ is the only evolutionarily stable strategy (ESS) in this case. However, if staying on the edge $\alpha = 1$ (complete

non-participation), the population will be under neutral drift, allowing it to reestablish cooperation with cooperativity $\beta > \beta^*$ and subsequently overcome the barrier of complete non-participation with $\alpha < 1$.

Lastly, if the return on investment $r > (2n-2)/(n-2) > 2$ and the payoff for non-participation $\sigma < r - 1$, the only ESS is $(0,1)$, although the game is no longer a social dilemma in this scenario.

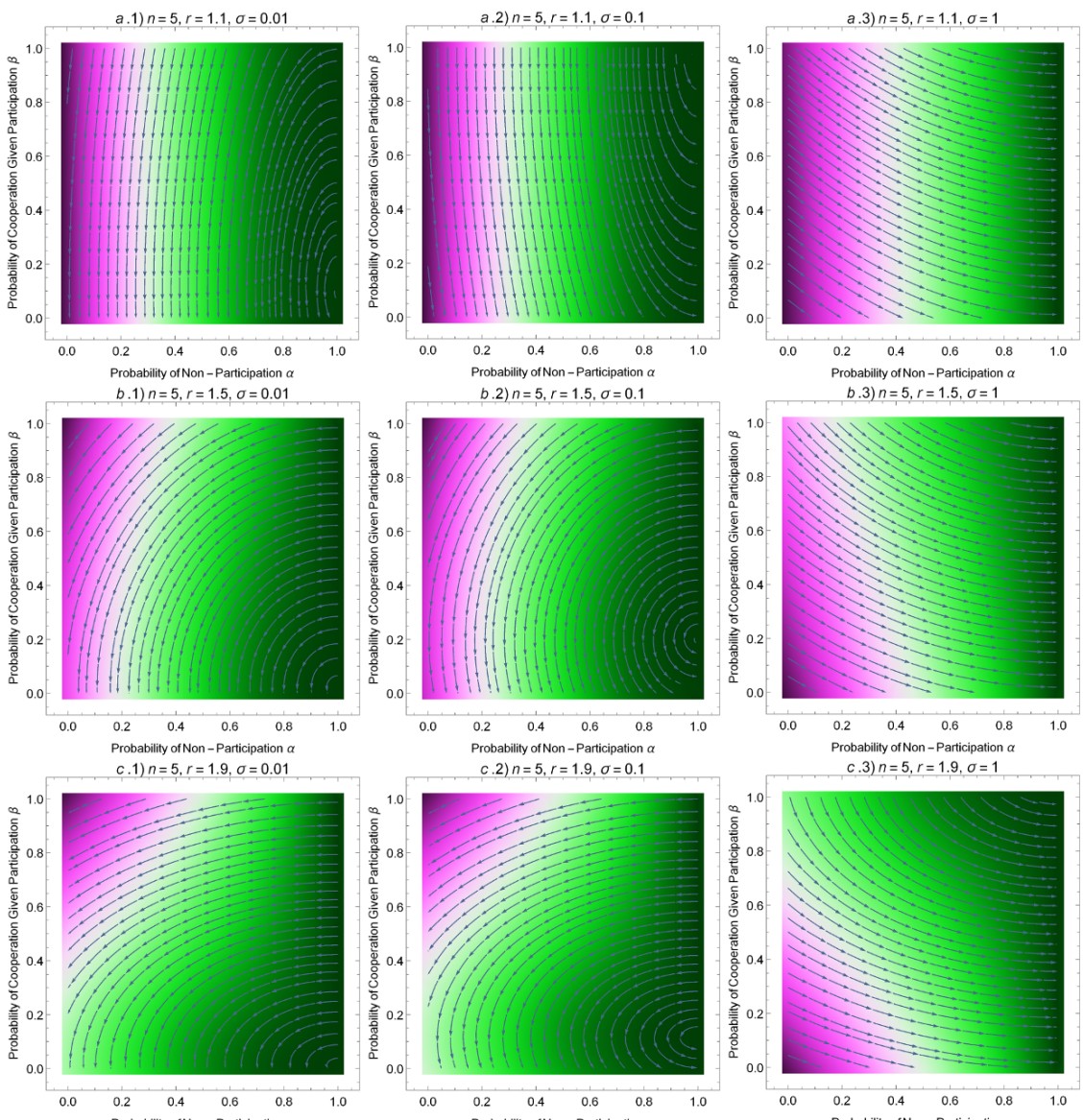

**Figure 4.** Coevolution of cooperation and stochastic opting-out. Shown in Panels (**a**–**c**) are the adaptive dynamics using the *StreamPlot* function of Mathematica in a finite population of size $n = 5$ for the selection pressure $\gamma = 1$ and various values of return on investment, $r$, and payoff for non-participants, $\sigma$. Following the arrows leads to the most likely path the population will take in the strategy space. Please note that if $\sigma < r - 1$, there exists a critical threshold of cooperativity $\beta^* = \sigma/(r-1)$ such that increasing likelihood of participation is more beneficial for agents with cooperativity $\beta > \sigma/(r-1)$ whereas participating cooperators are always prone to exploitation by others as shown in Panels (**b.2**) and (**c.2**). Hence, for $r < (2n-2)/(n-2)$, the only evolutionarily stable strategy (ESS) is $(1,0)$. However, if $r > (2n-2)/(n-2) > 2$ where the game is no longer a social dilemma, $(0,1)$ is an ESS.

## 4. Discussion and Conclusions

In this work, we study and quantify the role of stochastic opting-out in the evolution of group cooperation in PGGs and derive the exact condition for natural selection to favor cooperation in finite populations. We find the threshold of return on investment $r$, denoted by $R(\alpha)$, above which cooperation is favored is monotonically decreasing with $\alpha$, suggesting that allowing prescribed probabilistic participation can facilitate the evolution of cooperation. In the two extreme cases, we find that $R(0) = N(n-1)/(n-N)$ and $R(1^-) = 2(n-1)/(n-2)$ where $n$ is the population size and $N$ is the PGG size. Therefore, in the limit of large populations, increasing the likelihood of opting-out $\alpha$ can greatly reduce the threshold from $N$ to 2. This limiting result helps us to intuitively understand the role of stochastic opting-out.

Moreover, it seems that the effect of allowing stochastic opting-out is solely reducing the effective PGG size to $N(1-\alpha)$ on average, and thus one may expect the critical threshold of $r$ to be $R_{exp} = N(1-\alpha)(n-1)/(n-N(1-\alpha))$. However, we show that $R(\alpha) > R_{exp}$ for some limiting cases (see Figure 3). Such discrepancy is largely owing to the stochastic nature of the opting-out, in which the PGG interaction groups are formed by sampling players in finite populations and thus can be of varying size. Complementing prior studies of how group size affects cooperation [22,45], our study provides analytical insights into understanding how stochastic opting out causes dynamic PGG size and its effect on the evolutionary dynamics of group cooperation in finite populations.

All other things equal, whether it pays for players to switch from defection to cooperation, if participating in the PGG, depends on the net return from one's own contribution, that is, $(r/S - 1)$, where $S$ is the actual PGG size which can be less than $N$ due to stochastic opting-out by other players. Averaging such payoff difference from switching, $(r/S - 1)$, over all possible PGG sizes and taking into account whether the focal individual is a cooperator or a defector, we can obtain the expected payoff difference, $\pi_c - \pi_d$, which is in fact does not depend on $\sigma$ but on $\alpha$, as given in Equation (4). Intuitively, stochastic non-participation can possibly lead to small PGG size $S$ such that the return on one's own cooperation $r/S - 1 > 0$ is positive, and as a result, cooperation can be promoted if the probability of non-participation $\alpha$ exceeds $\alpha_0$, such that the resulting PGG size $S$ is likely to be sufficiently small to sustain cooperation.

Here we consider the simplest possible opting-out behavior, that is, random non-participation with a given probability $\alpha$. As such, every player, either a cooperator or a defector, has the same prescribed probability to abstain from the PGG. It is possible that the choice of opting-out may be endogenously made by players with their knowledge of the composition of the population. Namely, players are able to choose whether or not to participate based on their sensing of their potential interaction groups, for example, via quorum sensing [5,23]. In addition, opting-out decision may be conditional on whether there will be sufficient number of cooperators in the group such that the payoff from participating in the PGG outweighs the payoff of non-participation. Clearly, cooperators should be more picky than defectors when deciding whether or not to participate, because of their risk of being exploited by defectors. It is likely that natural selection will favor conditional non-participation strategies, for example, in scenarios qualitatively similar to what our adaptive dynamics analysis (Equation (35)) has revealed: only if the cooperativity in the population is sufficient high, $\beta > \sigma/(r-1)$, does it pay to participate. Therefore, it is of interest for future work to explore how these conditional opting-out behavior emerge in the first place and its impact on the long-term evolution of cooperation.

In conclusion, we find that in situations where samples of individuals are repeatedly drawn from a population for participation in PGGs, allowing for the possibility that members of a population stochastically cannot participate in the game facilitates cooperation. Furthermore, adaptive dynamics suggests that in the presence of small and minute mutations, introducing stochastic non-participation slows the rate at which the population tends to defection. While the adaptive dynamics also suggests that the population must tend to complete non-participation (i.e., $\alpha = 1$) when $r < (2n-2)/(n-2)$, although we may see brief bursts of cooperation arising from the upper part of the edge $\alpha = 1$ with $\beta > \sigma/(r-1)$ due to neutral drift. Additionally, using the

pairwise comparison updating rule [43,46], our results are valid both for games with behavioral strategies and games with genetic strategies. Since PGGs are also found widely in nature, (see Nadell et al. [5], Goryunov [6], Melis and Semmann [9] as examples), our results shed light on the evolution of cooperation in many biological and social situations [2,7,8,19,24,45].

**Author Contributions:** A.G.G. & F.F. conceived the model, A.G.G. analyzed the model with contributions from F.F., and A.G.G. & F.F. wrote the manuscript.

**Funding:** This research is supported by the Dartmouth Startup Fund, the Walter and Constance Burke Research Initiation Award, and a Junior Faculty Fellowship to F.F.

**Acknowledgments:** The authors would like to thank the National Science Foundation and Dartmouth College for funding the REU program at which the research was conducted. In particular, A.G.G. would also like to thank Anne Gelb, Tracy Moloney, and Amy Powell, all of Dartmouth College, for personally overseeing the program. Lastly, A.G.G. would like to thank Ignacio Uriarte-Tuero, George Pappas, Tsvetanka Tsendova, and Teena Gerhardt, all of Michigan State University, for making sure he attended a program that fit his needs.

**Conflicts of Interest:** We have no competing interests. The funders had no role in the design of the study; in the collection, analyses, or interpretation of data; in the writing of the manuscript, and in the decision to publish the results.

## Appendix A. Derivation of $\pi_d$

We define the probability that an event $E$ occurs be denoted by $P(E)$, and let the probability that $E$ occurs given a second event $F$ occurs be denoted by $P(E|F)$. Then,

$$\pi_d = \alpha\sigma + \Sigma P(n_c \cap S \cap plays) * payoff \tag{A1}$$

$$= \alpha\sigma + \Sigma P(plays)P(S|plays)P(n_c|S \cap plays) * payoff \tag{A2}$$

$$= \alpha\sigma + (1-\alpha)\Sigma P(S|plays)P(n_c|S \cap plays) * payoff \tag{A3}$$

$$= \alpha\sigma + (1-\alpha)[\Sigma_{S=2}^{N}P(S|plays)\Sigma_{n_c=0}^{S-1}P(n_c|S \cap plays)rn_c/S + P(S=1|plays)\sigma] \tag{A4}$$

$$= \alpha\sigma + (1-\alpha)[\Sigma_{S=2}^{N}P(S|plays)/S\Sigma_{n_c=0}^{S-1}P(n_c|S \cap plays)rn_c + P(S=1|plays)\sigma]. \tag{A5}$$

Substituting the values of the desired probabilities and simplifying,

$$\pi_d = \alpha\sigma + (1-\alpha)[r\Sigma_{S=2}^{N}(1/S)\binom{N-1}{S-1}(1-\alpha)^{S-1}\alpha^{N-S}\Sigma_{n_c=0}^{S-1}\binom{S-1}{n_c}* \tag{A6}$$

$$(x_c n/(n-1))^{n_c}(1-(x_c n/(n-1)))^{S-1-n_c}n_c + \alpha^{N-1}\sigma]$$

$$= \alpha\sigma + (1-\alpha)[r\Sigma_{S=2}^{N}\binom{N-1}{S-1}(1-\alpha)^{S-1}\alpha^{N-S}(x_c n/(n-1))(S-1)/S\Sigma_{n_c=1}^{S-1}\binom{S-2}{n_c-1}* \tag{A7}$$

$$(x_c n/(n-1))^{n_c-1}(1-(x_c n/(n-1)))^{S-1-n_c} + \alpha^{N-1}\sigma]$$

$$= \alpha\sigma + (1-\alpha)[r\Sigma_{S=2}^{N}\binom{N-1}{S-1}(1-\alpha)^{S-1}\alpha^{N-S}(x_c n/(n-1))(S-1)/S\Sigma_{k=0}^{S-2}\binom{S-2}{k}* \tag{A8}$$

$$(x_c n/(n-1))^{k}(1-(x_c n/(n-1)))^{S-2-k} + \alpha^{N-1}\sigma].$$

Continuing to simplify,

$$\pi_d = \alpha\sigma + (1-\alpha)[r\Sigma_{S=2}^N \binom{N-1}{S-1}(1-\alpha)^{S-1}\alpha^{N-S}(x_c n/(n-1))(S-1)/S + \alpha^{N-1}\sigma] \tag{A9}$$

$$= \alpha\sigma + (1-\alpha)[r(x_c n/(n-1))(\Sigma_{S=2}^N \binom{N-1}{S-1}(1-\alpha)^{S-1}\alpha^{N-S} - \Sigma_{S=2}^N \binom{N-1}{S-1}(1-\alpha)^{S-1}*$$
$$\alpha^{N-S}/S) + \alpha^{N-1}\sigma] \tag{A10}$$

$$= \alpha\sigma + (1-\alpha)[r(x_c n/(n-1))(\Sigma_{k=1}^{N-1}\binom{N-1}{k}(1-\alpha)^k\alpha^{N-k-1} - (1/N)\Sigma_{S=2}^N\binom{N}{S}(1-\alpha)^{S-1}*$$
$$\alpha^{N-S}) + \alpha^{N-1}\sigma] \tag{A11}$$

$$= \alpha\sigma + (1-\alpha)[r(x_c n/(n-1))((1-\alpha^{N-1}) - 1/(N(1-\alpha))\Sigma_{S=2}^N\binom{N}{S}(1-\alpha)^S\alpha^{N-S}) + \alpha^{N-1}\sigma] \tag{A12}$$

$$= \alpha\sigma + (1-\alpha)[r(x_c n/(n-1))((1-\alpha^{N-1}) - [1-N(1-\alpha)\alpha^{N-1}-\alpha^N]/[N(1-\alpha)] + \alpha^{N-1}\sigma] \tag{A13}$$

$$= \alpha\sigma + (1-\alpha)[r(x_c n/(n-1))(N-N\alpha-[1-\alpha^N])/(N[1-\alpha]) + \alpha^{N-1}\sigma] \tag{A14}$$

$$= \alpha\sigma + (1-\alpha)[r(x_c n/(n-1))[1-(1-\alpha^N)/[N(1-\alpha)]] + \alpha^{N-1}\sigma]. \tag{A15}$$

We have verified via Mathematica and via Hauert et al. [15] that

$$\Sigma_{S=2}^N P(S|plays)\Sigma_{n_c=0}^{N-1}P(n_c|S\cap plays)rn_c/S + P(S=1|plays)\sigma =$$
$$r(x_c n/(n-1))[1-(1-\alpha^N)/[N(1-\alpha)]] + \alpha^{N-1}\sigma. \tag{A16}$$

**Appendix B. Derivation of $\pi_c$**

$$\pi_c = \alpha\sigma + \Sigma P(n_c \cap S \cap plays) * payoff \tag{A17}$$

$$= \alpha\sigma + (1-\alpha)\Sigma P(S|plays)P(n_c|S\cap plays) * payoff \tag{A18}$$

$$= \alpha\sigma + (1-\alpha)[\Sigma_{S=2}^N P(S|plays)\Sigma_{n_c=0}^{S-1}P(n_c|S\cap plays)((r/S)(n_c+1)-1) +$$
$$P(S=1|plays)\sigma] \tag{A19}$$

$$= \alpha\sigma + (1-\alpha)[\Sigma_{S=2}^N P(S|plays)\Sigma_{n_c=0}^{S-1}P(n_c|S\cap plays)rn_c/S +$$
$$\alpha^{N-1}\sigma + \Sigma_{S=2}^N P(S|plays)\Sigma_{n_c=0}^{S-1}P(n_c|S\cap plays)(r/S-1)]. \tag{A20}$$

Please note that

$$\alpha\sigma + (1-\alpha)[\Sigma_{S=2}^N P(S|plays)\Sigma_{n_c=0}^{S-1}P(n_c|S\cap plays)rn_c/S + \alpha^{N-1}\sigma] = \pi_d, \tag{A21}$$

where $x_c n/(n-1)$ is replaced by $(x_c n-1)/(n-1)$. Indeed, $x_c n/(n-1)$ is the probability that if a player defects then another player will cooperate. Here, however, we consider the probability that if a *cooperator* plays then a player will cooperate. Thus, to account for the cooperator we know to be playing, we must consider $(x_c n-1)/(n-1)$. It follows that

$$\pi_c = \pi_d - (1-\alpha)r/(n-1)[1-(1-\alpha^N)/[N(1-\alpha)]] + (1-\alpha)\Sigma_{S=2}^N P(S|plays)*$$
$$\Sigma_{n_c=0}^{S-1}P(n_c|S\cap plays)(r/S-1) \tag{A22}$$

$$= \pi_d - (1-\alpha)r/(n-1)[1-(1-\alpha^N)/[N(1-\alpha)]] + (1-\alpha)\Sigma_{S=2}^N P(S|plays)(r/S-1) \tag{A23}$$

$$= \pi_d - (1-\alpha)r/(n-1)[1-(1-\alpha^N)/[N(1-\alpha)]] + (1-\alpha)*$$
$$[r\Sigma_{S=2}^N(1/S)\binom{N-1}{S-1}(1-\alpha)^{S-1}\alpha^{N-S} - \Sigma_{S=2}^N\binom{N-1}{S-1}(1-\alpha)^{S-1}\alpha^{N-S}]. \tag{A24}$$

Continuing to simplify,

$$
\pi_c = \pi_d - (1-\alpha)r/(n-1)[1 - (1-\alpha^N)/[N(1-\alpha)]] + (1-\alpha)*
$$
$$
[(r/N)\Sigma_{S=2}^N \binom{N}{S}(1-\alpha)^{S-1}\alpha^{N-S} - \Sigma_{k=1}^{N-1}\binom{N-1}{k}(1-\alpha)^k\alpha^{N-k-1}] \tag{A25}
$$

$$
= \pi_d - (1-\alpha)r/(n-1)[1 - (1-\alpha^N)/[N(1-\alpha)]] + (1-\alpha)[(r/[N(1-\alpha)])*
$$
$$
\Sigma_{S=2}^N \binom{N}{S}(1-\alpha)^S\alpha^{N-S} - (1-\alpha^{N-1})] \tag{A26}
$$

$$
= \pi_d - (1-\alpha)r/(n-1)[1 - (1-\alpha^N)/[N(1-\alpha)]] + (1-\alpha)[r/[N(1-\alpha)]*
$$
$$
(1 - N(1-\alpha)\alpha^{N-1} - \alpha^N) - 1 + \alpha^{N-1}] \tag{A27}
$$

$$
= \pi_d - (1-\alpha)r/(n-1)[1 - (1-\alpha^N)/[N(1-\alpha)]] + (1-\alpha)[-1 - r\alpha^{N-1}
$$
$$
+ \alpha^{N-1} + (r/N)(1-\alpha^N)/(1-\alpha)] \tag{A28}
$$

$$
= \pi_d - r/(n-1)[1 - \alpha - (1-\alpha^N)/N] + (1-\alpha)[-1 + (1-r)\alpha^{N-1} +
$$
$$
(r/N)(1-\alpha^N)/(1-\alpha)]. \tag{A29}
$$

Again, we have verified via Mathematica and via Hauert et al. [15] that

$$
\Sigma_{S=2}^N P(S|plays)\Sigma_{n_c=0}^{S-1}P(n_c|S\cap plays)(r/S-1) = \tag{A30}
$$
$$
-1 + (1-r)\alpha^{N-1} + (r/N)(1-\alpha^N)/(1-\alpha). \tag{A31}
$$

## Appendix C. Transition Matrix

We define $P$ be the transition matrix for the Markov chain formed by repeatedly iterating pairwise comparison. Then, $P_{i,i-1} = p_{cd}i(n-i)/[n(n-1)]$, and $P_{i,i+1} = p_{dc}i(n-i)/[n(n-1)]$, for $i$ = 2, 3, ..., $n-1$. Since the only other transition from i cooperators per iteration is the absence of transition, $P_{i,i} = 1 - P_{i,i-1} - P_{i,i+1}$, and the remaining entries in the $i^{th}$ row are 0. Also considering that $i = 0$ cooperators and $i = n$ cooperators are absorbing states, it follows that P is the tridiagonal $(n+1) \times (n+1)$ matrix

$$
\begin{bmatrix}
1 & 0 & 0 & 0 & \cdots & 0 \\
P_{2,1} & P_{2,2} & P_{2,3} & 0 & \cdots & 0 \\
0 & P_{3,2} & P_{3,3} & P_{3,4} & \ddots & \vdots \\
\vdots & \ddots & \ddots & \ddots & \ddots & 0 \\
0 & \cdots & 0 & P_{n-1,n-2} & P_{n-1,n-1} & P_{n-1,n} \\
0 & \cdots & 0 & 0 & 0 & 1
\end{bmatrix}. \tag{A32}
$$

Fortunately, the calculation $P^k$ as $k \to \infty$ is relatively straightforward. Indeed, the calculated the fixation probabilities $x_i$ in Equation (11), and $y_{n-i}$ in Equation (13), represent, respectively, the last and first entries in the $i^{th}$ row of $\lim_{n\to\infty} P^n$. Also considering that the entries in any given row of $P^n$ must sum to 1 as $P^n$ is a stochastic matrix, and that $x_i + y_{n-i} = 1$, it follows that

$$
\lim_{n\to\infty} P^n =
\begin{bmatrix}
1 & 0 & \cdots & 0 & 0 \\
x_1 & 0 & \cdots & 0 & y_{n-1} \\
x_2 & 0 & \cdots & 0 & y_{n-2} \\
\vdots & \vdots & \vdots & \vdots & \vdots \\
x_{n-1} & 0 & \cdots & 0 & y_1 \\
0 & 0 & \cdots & 0 & 1
\end{bmatrix}. \tag{A33}
$$

Thus, $\lim_{n\to\infty} XP^n$ converges to a vector of the form $(a, 0, ..., 0, b)$. Namely, the set of vectors of the form $(a, 0, \ldots, 0, b)$ is the set of eigenvectors of $\lim_{n\to\infty} P^n$, which in turn is the set of eigenvectors

of $P$ with eigenvalue 1. Moreover, if $X = (Prob(i = 0), Prob(i = 1), ..., Prob(i = n))$, then $\lim_{n \to \infty} XP^n$ converges to a vector of the form $(\alpha, 0, ..., 0, \beta)$, where $\alpha + \beta = 1$. Since the set of vectors of the form $(\alpha, 0, ..., 0, \beta)$ with $\alpha + \beta = 1$ is the set of stochastic eigenvectors of $P$ with eigenvalue 1, it follows that depending on the initial probability vector for the system, $X = (Prob(i = 0), Prob(i = 1), ..., Prob(i = n))$, the system can potentially converge to any stochastic eigenvector.

## Appendix D. Inequalities

*Appendix D.1. Proof of (15)*

$$
\begin{aligned}
&x_1 > 1/n \Leftrightarrow \\
&[1 - G]/[1 - G^n] > 1/n \Leftrightarrow \\
&[1 - G^n]/[1 - G] < n \Leftrightarrow \\
&\Sigma_{k=0}^{n-1} G^k < \Sigma_{k=0}^{n-1} 1 \Leftrightarrow \\
&G < 1. \quad \square
\end{aligned}
\tag{A34}
$$

*Appendix D.2. Proof of (16)*

$$y_1 < 1/n \Leftrightarrow \tag{A35}$$
$$[G^{n-1} - G^n]/[1 - G^n] < 1/n \Leftrightarrow \tag{A36}$$
$$[1 - G]/[1 - G^n] < 1/(nG^{n-1}) \Leftrightarrow \tag{A37}$$
$$[1 - G^n]/[1 - G] > nG^{n-1} \Leftrightarrow \tag{A38}$$
$$(1/n)\Sigma_{k=0}^{n-1} G^k > G^{n-1} \Leftrightarrow \tag{A39}$$
$$(1/n)\Sigma_{k=0}^{n-1} G^k > (G^{(n-1)(n)/2})^{2/n} \Leftrightarrow \tag{A40}$$
$$(1/n)\Sigma_{k=0}^{n-1} G^k > ((\Pi_{k=0}^{n-1} G^k)^{1/n})^2. \tag{A41}$$

Moreover, if $G < 1$, then $(\Pi_{k=0}^{n-1} G^k)^{\frac{1}{n}} > ((\Pi_{k=0}^{n-1} G^k)^{\frac{1}{n}})^2$. Hence, if $G < 1$, applying the arithmetic-mean-geometric-mean (AM-GM) inequality demonstrates that

$$(1/n)\Sigma_{k=0}^{n-1} G^k > ((\Pi_{k=0}^{n-1} G^k)^{1/n})^2. \tag{A42}$$

Thus, $G < 1$ implies that $y_1 < 1/n$. $\square$

*Appendix D.3. Proof of (17)*

If $G > 1$ and $n > 1$, note that

$$y_1 > 1/n \Leftrightarrow \tag{A43}$$
$$[G^{n-1} - G^n]/[1 - G^n] > 1/n \Leftrightarrow \tag{A44}$$
$$\frac{G - 1}{G - 1/G^{n-1}} > 1/n \Leftrightarrow \tag{A45}$$
$$G - G^{1-n} < nG - n. \tag{A46}$$

Then, observe that

$$\frac{d^2}{d^2 G}(G - G^{1-n}) = -n(n-1)G^{-n-1} < 0, \tag{A47}$$

for $n > 1$. Thus,

$$\frac{d}{dG}(G - G^{1-n}) = 1 - (1 - n)G^{-n} \qquad (A48)$$

is decreasing whereas

$$\frac{d}{dG}(nG - n) = n \qquad (A49)$$

is constant. Also considering that

$$\frac{d}{dG}(G - G^{1-n})|_{G \to 1} = n = \frac{d}{dG}(nG - n)|_{G \to 1}, \qquad (A50)$$

it follows that

$$\frac{d}{dG}(G - G^{1-n}) < \frac{d}{dG}(nG - n), \qquad (A51)$$

for $n > 1$. Since it is also true that as $G \to 1$, $G - G^{n-1} \to 0$ and $nG - n \to 0$,

$$G - G^{1-n} < nG - n. \qquad (A52)$$

Therefore, if $G > 1$, $y_1 > 1/n$.  □

*Appendix D.4. Lemma 1:* $1 - \alpha^{N-1}N + \alpha^N(N - 1) > 0$

For $\alpha = 0$, $1 - \alpha^{N-1}N + \alpha^N(N - 1) = 1$. Then, if $\alpha \in (0, 1)$ and $N > 1$,

$$1 - \alpha^{N-1}N + \alpha^N(N - 1) > 0 \Leftrightarrow \qquad (A53)$$
$$[1 - \alpha^N]/[N(1 - \alpha)] - \alpha^{N-1} > 0 \Leftrightarrow \qquad (A54)$$
$$(1/N)(\Sigma_{k=0}^{N-1}\alpha^k) - \alpha^{N-1} > 0 \Leftrightarrow \qquad (A55)$$

$$(1/N)(\Sigma_{k=0}^{N-1}\alpha^k) > (\alpha^{(N-1)N/2})^{\frac{2}{N}} \Leftrightarrow \qquad (A56)$$
$$(1/N)(\Sigma_{k=0}^{N-1}\alpha^k) > ((\Pi_{k=0}^{N-1}\alpha^k)^{1/N})^2. \qquad (A57)$$

However, since $((\Pi_{k=0}^{N-1}\alpha^k)^{1/N})^2 = \alpha^{N-1} < 1$,

$$((\Pi_{k=0}^{N-1}\alpha^k)^{1/N})^2 < ((\Pi_{k=0}^{N-1}\alpha^k)^{1/N}), \qquad (A58)$$

and since by the AM-GM inequality,

$$(1/N)(\Sigma_{k=0}^{k=N-1}\alpha^k) > ((\Pi_{k=0}^{N-1}\alpha^k)^{1/N}) \qquad (A59)$$

The inequality $1 - \alpha^{N-1}N + \alpha^N(N - 1) > 0$ must be valid.  □

*Appendix D.5. Lemma 2:* $1 - [1 - \alpha^N]/[N(1 - \alpha)] > 0$

Suppose $\alpha \in [0, 1)$. Then,

$$[1 - \alpha^N]/[N(1 - \alpha)] = 1/N\Sigma_{k=0}^{N-1}\alpha^k \qquad (A60)$$
$$< 1/N\Sigma_{k=0}^{N-1}1 \qquad (A61)$$
$$< 1 \qquad (A62)$$

Hence, $1 - [1 - \alpha^N]/[N(1 - \alpha)] > 0$. □

*Appendix D.6. Proof that as $N/n \to 0$, $R(\alpha) > N(1-\alpha) \approx R_{exp}(\alpha)$*

As $N/n \to 0$, $R(\alpha) \to N\dfrac{1 - \alpha - \alpha^{N-1} + \alpha^N}{1 - \alpha^{N-1}N + \alpha^N(N-1)}$. Hence,

$$N(1-\alpha) < N\frac{1 - \alpha - \alpha^{N-1} + \alpha^N}{1 - \alpha^{N-1}N + \alpha^N(N-1)} \Leftrightarrow \tag{A63}$$

$$(1-\alpha)(1 - \alpha^{N-1}N + \alpha^N(N-1)) < 1 - \alpha - \alpha^{N-1} + \alpha^N \Leftrightarrow \tag{A64}$$

$$1 - \alpha^{N-1}N + \alpha^N(N-1) < 1 - \alpha^{N-1} \Leftrightarrow \tag{A65}$$

$$\alpha^N(N-1) < \alpha^{N-1}(N-1) \Leftrightarrow \tag{A66}$$

$$\alpha^N < \alpha^{N-1}, \tag{A67}$$

which is true for $\alpha \in (0,1)$. Therefore, the inequality (A63) holds if and only if the denominator of the right-hand-side of (A63) is positive. This is true by Lemma 1. Additionally, as $N/n \to 0$, $n - N(1-\alpha) \to n \approx n - 1$, so

$$R_{exp}(\alpha) = \frac{(n-1)N(1-\alpha)}{n - N(1-\alpha)} \tag{A68}$$

$$= N(1-\alpha). \tag{A69}$$

□

*Appendix D.7. Lemma 3: Behavior of $R(\alpha)$ as $\alpha \to 0$*

As $\alpha \to 0$,

$$R(\alpha) = N\frac{1 - \alpha - \alpha^{N-1} + \alpha^N}{1 - \dfrac{N-1}{n-1} + \dfrac{N}{n-1}\alpha - N\alpha^{N-1} + \alpha^N(N-1-\dfrac{1}{n-1})} \tag{A70}$$

$$\to \frac{N}{1 - (N-1)/(n-1)} \tag{A71}$$

$$= \frac{N(n-1)}{n - N} \tag{A72}$$

$$> 0, \tag{A73}$$

provided that $N \neq n$. On the other hand, if $N = n$, then,

$$R(\alpha) = N\frac{1 - \alpha - \alpha^{N-1} + \alpha^N}{\dfrac{N}{n-1}\alpha - N\alpha^{N-1} + \alpha^N(N-1-\dfrac{1}{n-1})} \tag{A74}$$

$$\to N\frac{1-\alpha}{N/(N-1)\alpha} \tag{A75}$$

$$\to (N-1)\frac{1-\alpha}{\alpha} \tag{A76}$$

$$\to +\infty. \tag{A77}$$

Hence, as $\alpha \to 0$, $R(\alpha)$ is positive.

## Appendix E. Proof that $R(\alpha)$ Is Strictly Decreasing on $[0,1)$

Let

$$F(\alpha) = r([1 - \alpha^N]/[N(1-\alpha)] - \alpha^{N-1}) - (1 - \alpha^{N-1}). \tag{A78}$$

As shown in Hauert et al. [15], $F$ on $(0,1)$ has no root for $r \leq 2$. The preceding result does not hold, though, if $N = 2$. We address the case for which $N = 2$ at the end of the following proof. For now, we suppose $N > 2$. Then, for every $r > 2$ there exists exactly one $\alpha$ such that $F = 0$, as shown in Hauert et al. [15]. We consider

$$Q(\alpha) = \frac{1 - \alpha^{N-1}}{[1 - \alpha^N]/[N(1 - \alpha)] - \alpha^{N-1}}. \tag{A79}$$

$Q$ gives the values of $r$ given $\alpha$ for which $F$ is zero. Hence, $Q$ is injective where it is defined. Since $[1 - \alpha^N]/[N(1 - \alpha)] - \alpha^{N-1}$ is positive on $[0,1)$ by Lemma 1 in Appendix D, $Q$ is defined and thus injective on $(0,1)$. Thus, $Q$ is either strictly decreasing or strictly increasing on $(0,1)$. However,

$$\lim_{\alpha \to 0} Q(\alpha) = N, \tag{A80}$$

and

$$\lim_{\alpha \to 1} Q(\alpha) = 2, \tag{A81}$$

applying L'Hospital's rule twice. Since $Q$ is continuous on $(0,1)$ there exist $\delta_1 < 1/2$ and $\delta_2 < 1/2$ such that for $\alpha_1 \in (0, 0 + \delta_1)$ and for $\alpha_2 \in (1 - \delta_2, 1)$, $|Q(\alpha_1) - N| < 1/3$ and $|Q(\alpha_2) - 2| < 1/3$, respectively. Choosing arbitrary $c_1 \in (0, 0 + \delta_1)$ and $c_2 \in (1 - \delta_2, 1)$, it follows that for $N > 2$, $Q(c_1) > Q(c_2)$ and $c_1 < c_2$. Hence, $Q$ must be strictly decreasing on $(0,1)$. Moreover, $Q(0) = N$. Also considering that $Q < N$ on $(0,1)$, which can be proven using Lemma 2, $Q$ is strictly decreasing on $[0,1)$.

Then, we let the numerator of $Q$ be d

$$S(\alpha) = 1 - \alpha^{N-1}, \tag{A82}$$

and note that $S$ is strictly decreasing but positive on $[0,1)$. Next, we let the denominator of $Q$ be

$$T(\alpha) = [1 - \alpha^N]/[N(1 - \alpha)] - \alpha^{N-1}, \tag{A83}$$

which is positive in (0,1) by Lemma 1 in Appendix D. Lastly, we let

$$U(\alpha) = -\frac{1}{n - 1}\left(1 - \frac{1 - \alpha^N}{N[1 - \alpha]}\right), \tag{A84}$$

which is negative on $(0,1)$ by Lemma 2. Also, $[1 - \alpha^{N-1}]/[1 - \alpha] = \Sigma_{k=0}^{N-2}\alpha^k$ for $N \geq 2$, a strictly increasing function of $\alpha$ for $\alpha \geq 0$ if $N > 2$ and constant if $N = 2$. Hence, $1 - [1 - \alpha^{N-1}]/[1 - \alpha]$ is strictly decreasing, and hence $U(\alpha) = -\frac{1}{n - 1}\left(1 - \frac{1 - \alpha^N}{N[1 - \alpha]}\right)$ is strictly increasing for $N \geq 2$ and constant for $N = 2$.

Next, note that

$$R(\alpha) = S(\alpha)/[T(\alpha) + U(\alpha)], \tag{A85}$$

and suppose for contradiction that $T(\alpha) + U(\alpha)$ has zeroes in $(0,1)$ which form some set $W$. Since $T(\alpha) + U(\alpha)$ is a polynomial, $W$ must be finite. Then, we may choose $w_0 = \min\{w \in W\}$. Thus, throughout the interval $(0, w_0)$, $T(\alpha) + U(\alpha)$ must be either positive or negative but not both. Moreover, since $S(\alpha)$ is positive on $(0,1)$, $R(\alpha)$ cannot change sign on $(0,1)$. Also considering that by Lemma 3, $R(\alpha)$ is positive as $\alpha \to 0$, it follows that $R(\alpha)$ and $T(\alpha) + U(\alpha)$ are positive on $(0, w_0)$. Next, consider any $\alpha_{inv}, \alpha \in (0, w_0)$ such that $\alpha_{inv} < \alpha$. Then,

$$R(\alpha_{inv}) > R(\alpha) \Leftrightarrow \tag{A86}$$

$$S(\alpha_{inv})/[T(\alpha_{inv}) + U(\alpha_{inv})] > S(\alpha)/[T(\alpha) + U(\alpha)] \Leftrightarrow \tag{A87}$$

$$S(\alpha_{inv})T(\alpha) + S(\alpha_{inv})U(\alpha) > S(\alpha)T(\alpha_{inv}) + S(\alpha)U(\alpha_{inv}). \tag{A88}$$

We can show that

$$S(\alpha_{inv})U(\alpha) > S(\alpha)U(\alpha_{inv}). \tag{A89}$$

Details of proof is deferred to the end of Appendix E.

Furthermore, since $Q$ is strictly decreasing and $T$ is positive,

$$S(\alpha_{inv})T(\alpha) > S(\alpha)T(\alpha_{inv}). \tag{A90}$$

Equation (A89) and Equation (A90) together imply that Equation (A88) is valid on $(0,1)$. Thus, $R$ is strictly decreasing on $(0, w_0)$ for $N > 2$. Moreover, since $S$ and $T(\alpha) + U(\alpha)$ are both polynomials, and since $S$ is nonzero on $(0,1)$, it follows that $R$ must have an asymptote at $w_0$. However, $R$ is positive and strictly decreasing on $(0, w_0)$, so $R$ cannot tend to $\pm\infty$ at $w_0$ and thus cannot have an asymptote at $w_0$. This is a contradiction! It must thus be false that $T(\alpha) + U(\alpha)$ has any zeroes on $(0,1)$. Hence, since $R$ and $S$ are both positive as $\alpha \to 0$, $T(\alpha) + U(\alpha)$ must also positive as $\alpha \to 0$. We may then show that $R(\alpha)$ is strictly decreasing on $(0,1)$ by applying an argument analogous to the argument used to show that $R(\alpha)$ was strictly decreasing on $(0, w_0)$ in the preceding proof by contradiction. Namely, we have already established that Equation (A88) holds on $(0,1)$. Since $T(\alpha) + U(\alpha)$ is positive on $(0,1)$ Equation (A88) still implies Equation (A86). Therefore, $R(\alpha)$ is strictly decreasing on $[0,1)$ for $N > 2$.

However, if $N = 2$, then the only change from the above proof is that $Q$ is constant rather than strictly decreasing. Then, Equation (A89) still holds, and we replace Equation (A90) by

$$S(\alpha_{inv})T(\alpha) = S(\alpha)T(\alpha_{inv}). \tag{A91}$$

Thus, Equation (A88) still holds. Hence, $R$ is strictly decreasing on $[0,1)$ for $N \geq 2$. $\square$

*Proof $S(\alpha_{inv})U(\alpha) > S(\alpha)U(\alpha_{inv})$ for $\alpha_{inv} < \alpha$*

It will now be very useful to show that

$$S(\alpha_{inv})U(\alpha) > S(\alpha)U(\alpha_{inv}). \tag{A92}$$

To demonstrate the preceding relations holds, note that since $S$ is strictly decreasing, $S(\alpha_{inv}) > S(\alpha)$. Also considering that $U$ is strictly increasing, $U(\alpha) > U(\alpha_{inv})$, and considering that $S$ is positive and $U$ is negative, it follows that

$$S(\alpha_{inv})U(\alpha) > S(\alpha)U(\alpha_{inv}) \Leftrightarrow \tag{A93}$$

$$S(\alpha_{inv})U(\alpha)/U(\alpha_{inv}) < S(\alpha) \Leftrightarrow \tag{A94}$$

$$S(\alpha_{inv})/U(\alpha_{inv}) > S(\alpha)/U(\alpha). \tag{A95}$$

In other words, $S(\alpha_{inv})U(\alpha) > S(\alpha)U(\alpha_{inv})$ if and only if $S(\alpha)/U(\alpha)$ is strictly decreasing. Indeed, to see why $S(\alpha)/U(\alpha)$ is strictly decreasing, observe that

$$S(\alpha)/U(\alpha) = (1 - \alpha^{N-1})/\left(\frac{-1}{n-1}\left[1 - \frac{1 - \alpha^N}{N(1-\alpha)}\right]\right) \tag{A96}$$

$$= -(n-1)(1 - \alpha^{N-1})/\left(1 - \frac{1 - \alpha^N}{N(1-\alpha)}\right) \tag{A97}$$

$$= -(n-1)(1 - \alpha^{N-1})/\frac{N(1-\alpha) - (1 - \alpha^N)}{N(1-\alpha)} \tag{A98}$$

$$= -N(n-1)\frac{1 - \alpha - \alpha^{N-1} + \alpha^N}{N - 1 - N\alpha + \alpha^N}. \tag{A99}$$

Also considering that since $S$ is positive but $U$ is negative, $-N(n-1)$ is negative, so $S(\alpha)/U(\alpha)$ is strictly decreasing if and only if

$$X(\alpha) = \frac{1 - \alpha - \alpha^{N-1} + \alpha^N}{N - 1 - N\alpha + \alpha^N} \tag{A100}$$

is strictly increasing. Now, to establish that $X(\alpha)$ is strictly increasing, we will show $X'(\alpha)$ is positive. Indeed, observe that

$$0 < X'(\alpha) \Leftrightarrow \tag{A101}$$

$$0 < [(-1 - [N-1]\alpha^{N-2} + N\alpha^{N-1})(N - 1 - N\alpha + \alpha^N) \\ - (1 - \alpha - \alpha^{N-1} + \alpha^N)(-N + N\alpha^{N-1})]/[N - 1 - N\alpha + \alpha^N]^2 \Leftrightarrow \tag{A102}$$

$$0 < [(-1 - [N-1]\alpha^{N-2} + N\alpha^{N-1})(N - 1 - N\alpha + \alpha^N) \\ - (1 - \alpha - \alpha^{N-1} + \alpha^N)(-N + N\alpha^{N-1})] \Leftrightarrow \tag{A103}$$

$$0 < -(N-1) + N\alpha - \alpha^N - (N-1)^2\alpha^{N-2} + N(N-1)\alpha^{N-1} - \\ (N-1)\alpha^{2N-2} + N(N-1)\alpha^{N-1} - N^2\alpha^N + N\alpha^{2N-1} - (-N+ \\ N\alpha + N\alpha^{N-1} - N\alpha^N + N\alpha^{N-1} - N\alpha^N - N\alpha^{2N-2} + N\alpha^{2N-1}) \Leftrightarrow \tag{A104}$$

$$0 < 1 + \alpha^N(-1 - N^2 + 2N) - \alpha^{N-2}(N-1)^2 + \alpha^{N-1}(N(N-1)+ \\ N(N-1) - 2N) + \alpha^{2N-2}(N - (N-1)) \Leftrightarrow \tag{A105}$$

$$0 < 1 - \alpha^N(N-1)^2 - \alpha^{N-2}(N-1)^2 + \alpha^{N-1}(2N^2 - 4N) + \alpha^{2N-2}. \tag{A106}$$

However, it is not at all clear that the preceding function, $X_1(\alpha) = 1 - \alpha^N(N-1)^2 - \alpha^{N-2}(N-1)^2 + \alpha^{N-1}(2N^2 - 4N) + \alpha^{2N-2}$, is positive on (0,1). Testing the endpoints, we see that $X_1(0) = 1$, but

$$X_1(1) = 1 - (N-1)^2 - (N-1)^2 + 2N^2 - 4N + 1 \tag{A107}$$
$$= 1 - 2N^2 + 4N - 2 + 2N^2 - 4N + 1 \tag{A108}$$
$$= 0. \tag{A109}$$

Thus, if we can show that $X_1(\alpha)$ is strictly decreasing on $(0,1)$, we have that $X_1(\alpha) > 0$ on $(0,1)$. To that end, we will attempt to show that $X_1'(\alpha) < 0$ on $(0,1)$:

$$0 > X_1'(\alpha) \Leftrightarrow \tag{A110}$$

$$0 > -\alpha^{N-1}N(N-1)^2 - \alpha^{N-3}(N-1)^2(N-2)+ \\ \alpha^{N-2}2N(N-1)(N-2) + 2(N-1)\alpha^{2N-3} \Leftrightarrow \tag{A111}$$

$$0 > -\alpha^2 N(N-1) - (N-1)(N-2) + \alpha 2N(N-2) + 2\alpha^N. \tag{A112}$$

Still, it is not clear whether the preceding function, $X_2(\alpha) = -\alpha^2 N(N-1) - (N-1)(N-2) + \alpha 2N(N-2) + 2\alpha^N$, is negative on $(0,1)$. Testing endpoints again, we see that $X_2(0) = -(N-1)(N-2) < 0$ since $N > 2$, and that

$$X_2(1) = -N(N-1) - (N-1)(N-2) + 2N(N-2) + 2 \tag{A113}$$
$$= -N^2 + N - N^2 + 3N - 2 + 2 \tag{A114}$$
$$= 0. \tag{A115}$$

Thus, if we can show that $X_2(\alpha)$ is strictly increasing on (0,1), we have that $X_2(\alpha) < 0$. To do so, observe that

$$0 < X_2'(\alpha) \Leftrightarrow \tag{A116}$$

$$0 < -2\alpha N(N-1) + 2N(N-2) + 2N\alpha^{N-1} \Leftrightarrow \tag{A117}$$

$$0 < -\alpha(N-1) + (N-2) + \alpha^{N-1}. \tag{A118}$$

Even still, we need to do more work to show the preceding equation, $X_3(\alpha) = -\alpha(N-1) + (N-2) + \alpha^{N-1}$, is positive. Namely, testing endpoints one last time, we see that $X_3(0) = N - 2 > 0$ and that $X_3(1) = -(N-1) + N - 2 + 1 = 0$. Therefore, if we can show that $X_3(\alpha)$ is strictly decreasing on $(0, 1)$, then $X_3(\alpha) > 0$. One last time, note that

$$X_3'(\alpha) = -(N-1) + (N-1)\alpha^{N-1} \tag{A119}$$

$$< 0. \tag{A120}$$

Thus, $X_3$ is positive on $(0, 1)$. Employing the logic outlined above, we then have $X_2'(\alpha) > 0$, so $X_2(\alpha) < 0$. This in turn implies that $X_1'(\alpha) < 0$, so $X_1(\alpha) > 0$. Hence, $X'(\alpha) > 0$ and $X(\alpha)$ is strictly increasing. Finally, we then have that $S(\alpha)/U(\alpha)$ is strictly decreasing, yielding the desired result that $S(\alpha_{inv})U(\alpha) > S(\alpha)U(\alpha_{inv})$.

## Appendix F. Justification of Approximations

*Appendix F.1. Approximation for $R(\alpha)$ as $N \to \infty$, $\frac{N}{n-1} = c$*

We let $c$ be a real number in [0,1]. As $N \to \infty$, $\alpha^{N-1}N$, $\alpha^N(N-1-N/(n-1))$, $\alpha^{N-1}$, $\alpha^N$, $\alpha^{N+1}$ $\to 0$, as long as $\alpha \not\to 1$. Hence, for $\alpha \not\to 1$,

$$R(\alpha) = N \frac{1 - \alpha - \alpha^{N-1} + \alpha^N}{1 - \frac{N-1}{n-1} + \frac{N}{n-1}\alpha - N\alpha^{N-1} + \alpha^N(N-1-\frac{1}{n-1})} \tag{A121}$$

$$\approx N(1-\alpha)/(1 + c - c\alpha). \tag{A122}$$

However, applying L'Hospital's rule twice yields $\lim_{\alpha \to 1} R(\alpha) = 0$, which is $\lim_{\alpha \to 1} N(1 - \alpha)/(1 - c + c\alpha)$. $\square$

*Appendix F.2. Approximation for $R(\alpha)$ for $n \gg N \gg 0$*

As $n \to \infty$, $N \to \infty$, $N/n \to 0$, for $\alpha \not\to 1$,

$$R(\alpha) = N \frac{1 - \alpha - \alpha^{N-1} + \alpha^N}{1 - \frac{N-1}{n-1} + \frac{N}{n-1}\alpha - N\alpha^{N-1} + \alpha^N(N-1-\frac{1}{n-1})} \tag{A123}$$

$$\approx N(1-\alpha). \tag{A124}$$

However, as in the preceding proof, applying L'Hospital's rule twice yields $\lim_{\alpha \to 1} R(\alpha) = 0$, which is $\lim_{\alpha \to 1} N(1 - \alpha)$. $\square$

*Appendix F.3. Approximation for $R(\alpha)$ as $N \to \infty$*

As $N \to \infty$, $N\alpha^{N-1}, N\alpha^N \to 0$ for $\alpha \nrightarrow 1$, so for $\alpha \nrightarrow 1$,

$$R(\alpha) = N\frac{1 - \alpha - \alpha^{N-1} + \alpha^N}{1 - \dfrac{N-1}{n-1} + \dfrac{N}{n-1}\alpha - N\alpha^{N-1} + \alpha^N(N - 1 - \dfrac{1}{n-1})} \tag{A125}$$

$$\to N\frac{1 - \alpha}{1 - \dfrac{N-1}{n-1} + \dfrac{N}{n-1}\alpha} \tag{A126}$$

$$\to N\frac{1 - \alpha}{1 - \dfrac{N}{n} + \dfrac{N}{n}\alpha} \tag{A127}$$

$$\to \frac{nN(1-\alpha)}{n - N(1-\alpha)} \tag{A128}$$

$$\to \frac{(n-1)N(1-\alpha)}{n - N(1-\alpha)} \tag{A129}$$

$$= R_{exp}(\alpha) \tag{A130}$$

Additionally as in the preceding two proofs, applying L'Hosptal's rule twice yields $\lim_{\alpha \to 1} R(\alpha) = 0$, which is $\lim_{\alpha \to 1} R_{exp}(\alpha)$. $\square$

*Appendix F.4. Approximation for $R(\alpha)$ as $N/n \to 0$ and $\alpha \to 1$*

As $N/n \to 0$, $(N-1)/(n-1), N/(n-1), 1/(n-1) \to 0$, thus

$$R(\alpha) = N\frac{1 - \alpha - \alpha^{N-1} + \alpha^N}{1 - \dfrac{N-1}{n-1} + \dfrac{N}{n-1}\alpha - N\alpha^{N-1} + \alpha^N(N - 1 - \dfrac{1}{n-1})} \tag{A131}$$

$$\to N\frac{1 - \alpha - \alpha^{N-1} + \alpha^N}{1 - N\alpha^{N-1} + \alpha^N(N-1)}. \tag{A132}$$

Then, as $\alpha \to 1$ we arrive at the indeterminate form $0/0$, and employ L'Hospital's rule for the case $N > 2$. Now,

$$R(\alpha) \to N\frac{1 - \alpha - \alpha^{N-1} + \alpha^N}{1 - N\alpha^{N-1} + \alpha^N(N-1)} \tag{A133}$$

$$\to N\frac{-1 - (N-1)\alpha^{N-2} + N\alpha^{N-1}}{-N(N-1)\alpha^{N-2} + \alpha^{N-1}(N-1)N}. \tag{A134}$$

However, this still yields $0/0$, so we apply again L'Hospital's rule, obtaining

$$R(\alpha) \to N\frac{-1 - (N-1)\alpha^{N-2} + N\alpha^{N-1}}{-N(N-1)\alpha^{N-2} + \alpha^{N-1}(N-1)N} \tag{A135}$$

$$\to N\frac{-(N-1)(N-2)\alpha^{N-3} + N(N-1)\alpha^{N-2}}{-N(N-1)(N-2)\alpha^{N-3} + \alpha^{N-2}(N-1)^2N} \tag{A136}$$

$$\to N\frac{-(N-1)(N-2) + N(N-1)}{-N(N-1)(N-2) + (N-1)^2N} \tag{A137}$$

$$= 2 \tag{A138}$$

On the other hand, if $N = 2$, then $R(\alpha) \to N(1-\alpha)^2/(1-\alpha)^2 = N = 2$. Hence, as $N/n \to 0$ and $\alpha \to 1$, $R(\alpha) \to 2$.

**Appendix G. Derivation of $\pi_y$**

The payoff matrix for a two person public goods game in which cooperators invest 1 unit which is then multiplied by *r* and distributed equally among all players is

$$
\begin{array}{c|cc}
 & c & d \\
\hline
c & r-1 & r/2-1 \\
d & r/2 & 0
\end{array}
\tag{A139}
$$

Then, we suppose that there are *i* invaders in a population of $n-i$ defenders, and that the remaining individuals all play the same strategy. We let the invader be one of the players invited to play in the two person PGG and call that player "player *A*". Next, we let $A_c$, $A_d$, $A_n$, and $A_n^c$ represent the events where player A cooperates, defects, does not participate, and participates, respectively. We suppose "player B" is the other individual invited to play. We let $B_c$, $B_d$, $B_n$ be the events where player B cooperates, defects, and does not participate, respectively. Lastly, we let $E'$ and $E$ be the events where Player *B* invades and defends respectively. Denoting the intersection of any two events *F* and *G* by *FG*, and the probability that an event *F* occurs by $p(F)$,

$$
\begin{aligned}
\pi_y =& (r-1)[p(A_c E' B_c) + p(A_c E B_c)] + (r/2-1)[p(A_c E' B_d) + p(A_c E B_d)] \\
&+ r/2[p(A_d E' B_c) + p(A_d E B_c)] + \sigma[p(A_n) + p(A_n^c E' B_n) + p(A_n^c E B_n)]
\end{aligned}
\tag{A140}
$$

$$
\begin{aligned}
=& (r-1)[p(A_c)p(E'|A_c)p(B_c|A_c E') + p(A_c)p(E|A_c)p(B_c|A_c E)] \\
&+ (r/2-1)[p(A_c)p(E'|A_c)p(B_d|A_c E') + p(A_c)p(E|A_c)p(B_d|A_c E)] \\
&+ r/2[p(A_d)p(E'|A_d)p(B_c|A_d E') + p(A_d)p(E|A_d)p(B_c|A_d E)] \\
&+ \sigma[p(A_n) + p(A_n^c)p(E'|A_n^c)p(B_n|A_n^c E') + p(A_n^c)p(E|A_n^c)p(B_n|A_n^c E)]
\end{aligned}
\tag{A141}
$$

$$
\begin{aligned}
=& (r-1)\beta'(1-\alpha')[p(E'|A_c)\beta'(1-\alpha') + p(E|A_c)\beta(1-\alpha)] \\
&+ (r/2-1)\beta'(1-\alpha')[p(E'|A_c)(1-\beta')(1-\alpha') + p(E|A_c)(1-\beta)(1-\alpha)] \\
&+ r/2(1-\beta')(1-\alpha')[p(E'|A_d)\beta'(1-\alpha') + p(E|A_d)\beta(1-\alpha)] \\
&+ \sigma[\alpha' + (1-\alpha')[p(E'|A_n^c)\alpha' + p(EE|A_n^c)\alpha]
\end{aligned}
\tag{A142}
$$

$$
\begin{aligned}
\pi_y =& (r-1)\beta'(1-\alpha')\Big[\frac{i-1}{n-1}\beta'(1-\alpha') + \frac{n-i}{n-1}\beta(1-\alpha)\Big] \\
&+ (r/2-1)\beta'(1-\alpha')\Big[\frac{i-1}{n-1}(1-\beta')(1-\alpha') + \frac{n-i}{n-1}(1-\beta)(1-\alpha)\Big] \\
&+ r/2(1-\beta')(1-\alpha')\Big[\frac{i-1}{n-1}\beta'(1-\alpha') + \frac{n-i}{n-1}\beta(1-\alpha)\Big] \\
&+ \sigma\Big[\alpha' + (1-\alpha')\Big[\frac{i-1}{n-1}\alpha' + \frac{n-i}{n-1}\alpha\Big] \\
=& \frac{n-i}{n-1}(1-\alpha')(1-\alpha)(r\beta/2 + r\beta'/2 - \beta') + \frac{i-1}{n-1}(1-\alpha')^2\beta'(r-1) \\
&+ \sigma\alpha' + \sigma(1-\alpha')\Big(\frac{n-i}{n-1}\alpha + \frac{i-1}{n-1}\alpha'\Big)
\end{aligned}
\tag{A143}
$$
$$
\tag{A144}
$$

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
