# Peer review of "Evolution of Cooperation in Public Goods Games with Stochastic Opting-Out"

_games, doi:10.3390/g10010001_

Round 1

Reviewer 1 Report

All comments have been appropiately addressed.

Reviewer 2 Report

Authors improved the manuscript following all main suggestions.

(Just fix the missing reference, i.e. '?' at line 77 page 3: signaling (?), and optional participation)

This manuscript is a resubmission of an earlier submission. The following is a list of the peer review reports and author responses from that submission.

Round 1

Reviewer 1 Report

see the attached report

Author Response

Report on games-384793
"Evolution of Cooperation in Public Goods Games with Stochastic Opting-Out"
The paper studies the evolution of cooperation in a standard public goods game (PGG) appended with a (stochastically-determined) probability of opting out of the game. If the cooperation is notoriously difficult to evolve in such collective dilemmas, it is shown that increasing the probability of not playing the game decreases the threshold returns on investing in the public good, returns which are a key driver of cooperation in PGG. Thus, besides the population size and the game (sample) size a novel factor affecting the threshold returns to cooperation is identified. Natural selection of cooperators is therefore favored if players’ exogenous probability of opting out of the game increases. However, in the long run, where various participation probabilities/strategies may emerge under adaptive dynamics, simulations show that the population will evolve towards non-participation (\alpha= 1) and complete defection (\beta= 0):
Whereas the result that non-participating into a game may actually increase the fraction of cooperation in the games played is interesting in itself and, to a certain extent, puzzling, I would raise the following points:
Response: We thank the referee for positive feedback and helpful comments. We have revised our manuscript exactly as suggested. See our detailed response below.
(i) in the short run , R(\alpha) -the minimum return on investment for which the profit of cooperation exceeds the profit of defection - is shown to be decreasing in \alpha. As this is the key threshold for the emergence of cooperation, the (economic) intuition behind the link between non-participating players and the returns to cooperation needs to be further elucidated.
Response: Thanks for this suggestion. We have added a paragraph explaining the economic intuition regarding the connection between non-participation and the return to cooperation in the revised Discussion section. See the revised manuscript (p.19, lines 367-376.
(ii) in the model the option of not participating is exogenous to the players’ choice: would non-participation still be favored by the long-run adaptive dynamics process if players were given the choice between not playing and playing the PGG game?
Response: We appreciate this helpful question. In our model we consider the simplest possible opting-out behavior, that is, every player, either a cooperator or a defector, has the same prescribed probability of non-participation. However, if choice of opting-out is endogenously made by players with the knowledge of their potential interaction groups, for example, through quorum sensing, it is likely that natural selection will favor conditional non-participation strategies, whether there will be sufficient number of cooperators in the group such that the payoff from participating in the PGG outweighs the payoff of non-participation and that cooperators should be more picky than defectors when deciding whether or not to participate. We have added a new paragraph to discuss this possibility of model extension based on the adaptive dynamics results we have derived. See the revised manuscript ( p. 19-20, lines 377-392).
(iii) the authors claim that there is no evolutionarily stable state in the adaptive dynamics process. Could one envisage other attractors - e.g. cycles - in the (\beta, \alpha) space, different from the rest point \alpha= 1?
Response: We thank the referee for this insightful question. We have identified the conditions for ESS in relation to our model parameters. For $\sigma < r -1$, if $r < (2n-2)/(n-2)$, the only ESS is $(1,0)$; otherwise if $r > (2n-2)/(n-2)$, the ESS is $(0,1)$. We have revised manuscript to reflect this result, and moreover, we have elaborated on the long-term temporal dynamics based on our theoretical analysis of adaptive dynamics in finite populations. See revised manuscript (p. 16-17, lines 315-347).

Reviewer 2 Report

p.p1 {margin: 0.0px 0.0px 0.0px 0.0px; font: 12.0px Helvetica; color: #000000; background-color: #ffffff} p.p2 {margin: 0.0px 0.0px 0.0px 0.0px; font: 12.0px Helvetica; color: #000000; background-color: #ffffff; min-height: 14.0px} p.p3 {margin: 0.0px 0.0px 0.0px 0.0px; font: 12.0px Arial; color: #000000; background-color: #ffffff} p.p4 {margin: 0.0px 0.0px 0.0px 0.0px; font: 12.0px Helvetica; color: #000000; -webkit-text-stroke: #660099; background-color: #ffffff} p.p5 {margin: 0.0px 0.0px 0.0px 0.0px; font: 12.0px Helvetica; color: #000000; -webkit-text-stroke: #660099; background-color: #ffffff; min-height: 14.0px} span.s1 {font: 12.0px Helvetica} span.s2 {font: 12.0px Arial; font-kerning: none} span.s3 {font-kerning: none} span.s4 {-webkit-text-stroke: 0px #000000} span.s5 {font: 12.0px Arial; font-kerning: none; color: #660099}

In this work, authors study the evolutionary dynamics of dilemma games introducing a novel element, i.e. the possibility to be unable to participate in a game. In particular, while classical evolutionary models consider that selected agents always take part into the game, here a stochastic parameter is introduced for controlling this choice.

In doing so, the investigation aims to evaluate the relation between a stochastic opting-out and the emergence of cooperation. The work is developed by mean of an analytical approach and the results are finally provided and discussed.

In general, I think that this work is interesting and original. Here, a list of points authors should address before to provide a final recommendation:

- In relation to the first part of Introduction, where authors discuss about cooperation among biological species, they might find interesting the following work: “The host-pathogen game: an evolutionary approach to biological competitions. Front.Phys 6(94), 2018”, as well as “Evolutionary dynamics of group formation, PLoS ONE 12(11) 2017” discussing the relation between the size of groups and evolution

-Further mechanisms for triggering cooperation in dilemma games are described here: "Statistical Physics and Computational Methods for Evolutionary Game Theory, Springer 2018"

- The dynamics of non-participation can be compared to two mechanisms. One related to heterogeneous networks, where it has been shown that heterogeneity triggers cooperative behavior (see Evolutionary dynamics of social dilemmas in structured heterogeneous populations, PNAS 103(9), 2006), and one related to dynamical networks where agents vary the amount of opponents (see The role of competitiveness in the Prisoner’s Dilemma, Computational Social Networks 2(15), 2015) and to random motion that leads to similar scenario (see Statistical Physics of the Spatial Prisoner’s Dilemma with Memory-aware agents, EPJ-B (89), 2016). It would be interesting to clarify the difference between the proposed model and those that can be somehow similar, even if generated by different approach/method.

- In model section, I suggest to refer to ‘agent’ that is much more general, and participant to identify those agents that take part into the interaction game at a given time step.

- I would describe the reason for not participating to the game only in Introduction and Discussion/conclusion, not in the model section that should focus only on the model dynamics and related parameters

- Looking at Figure 1, the dynamics show different points of contact (at high level) with the scenario considered in “Statistical Physics of the Spatial Prisoner’s Dilemma with Memory-aware agents, EPJ-B (89), 2016”, where it has been investigated the role of random walk in the emergence of cooperation with different cluster of agents

- In Figure 1, I would use arrows for showing the motion of agents, while some labels for indicating the time flow (e.g. a), b) c)). In addition, I would extend a bit the caption that, in the current form, is poorly informative.

- It seems that sign “=“ is missing in eq.(6) and (8)

- In eq. (9) index of summation are missing in the inner part of the equation (I.e. the involved probabilities)

- Please, check eq.(22)

- The quality of figure (2) could be improved. In addition, now \gamma is defined as Selection Strength. Why?

- Please, check lines 139 and 140, maybe a part of the sentence has been removed during the generation of the file.

- Figure 4, the name of axis cannot be read. The resolution should be higher.

- In final discussion, I would describe the relation between \alpha and \sigma in relation to the results obtained in the investigation. These two parameters are relevant for the proposed model, so their relation could be interesting.

- It would be interesting a brief discussion/mention about the size of the population, with related results in this model, and the outcomes described in “Evolutionary dynamics of group formation, PLoS ONE 12(11) 2017” where population size has a strong relation with the evolutionary dynamics of a strategy

- Conclusion should be extended, including a wider discussion about the motivations, brief mention about the general model setup and approach to investigate it, and further discussion about points of contact with other fields and related open problems (e.g. biology, ecology, social systems, etc). In principle, Discussion and Conclusion could be merged in only one final section

Author Response

In this work, authors study the evolutionary dynamics of dilemma games introducing a novel element, i.e. the possibility to be unable to participate in a game. In particular, while classical evolutionary models consider that selected agents always take part into the game, here a stochastic parameter is introduced for controlling this choice.
In doing so, the investigation aims to evaluate the relation between a stochastic optingout and the emergence of cooperation. The work is developed by mean of an analytical approach and the results are finally provided and discussed.
In general, I think that this work is interesting and original. Here, a list of points authors should address before to provide a final recommendation:
Response: We are pleased to see the referee thinks our work is interesting and original. We have made an effort to thoroughly revised our manuscript and we hope these revisions have addressed your comments to your satisfaction.
- In relation to the first part of Introduction, where authors discuss about cooperation among biological species, they might find interesting the following work: “The hostpathogen game: an evolutionary approach to biological competitions. Front.Phys 6(94), 2018”, as well as “Evolutionary dynamics of group formation, PLoS ONE 12(11) 2017” discussing the relation between the size of groups and evolution

-Further mechanisms for triggering cooperation in dilemma games are described here: "Statistical Physics and Computational Methods for Evolutionary Game Theory, Springer 2018"
- The dynamics of non-participation can be compared to two mechanisms. One related to heterogeneous networks, where it has been shown that heterogeneity triggers cooperative behavior (see Evolutionary dynamics of social dilemmas in structured heterogeneous populations, PNAS 103(9), 2006), and one related to dynamical networks where agents vary the amount of opponents (see The role of competitiveness in the Prisoner’s Dilemma, Computational Social Networks 2(15), 2015) and to random motion that leads to similar scenario (see Statistical Physics of the Spatial Prisoner’s Dilemma with Memory-aware agents, EPJ-B (89), 2016). It would be interesting to clarify the difference between the proposed model and those that can be somehow similar, even if generated by different approach/method.
Response: We have cited and acknowledged these papers in our revised manuscript where appropriate. Thank you for bringing these papers to our attention.
- In model section, I suggest to refer to ‘agent’ that is much more general, and
participant to identify those agents that take part into the interaction game at a given time step.
Response: In response to this suggestion, we have revised the manuscript accordingly (p.4-5, lines 102-144).
- I would describe the reason for not participating to the game only in Introduction and Discussion/conclusion, not in the model section that should focus only on the model dynamics and related parameters
Response: We have streamlined the manuscript according to this helpful suggestion (see p.4-5, lines 102-144).
- Looking at Figure 1, the dynamics show different points of contact (at high level) with the scenario considered in “Statistical Physics of the Spatial Prisoner’s Dilemma with Memory-aware agents, EPJ-B (89), 2016”, where it has been investigated the role of random walk in the emergence of cooperation with different cluster of agents
- In Figure 1, I would use arrows for showing the motion of agents, while some labels for indicating the time flow (e.g. a), b) c)). In addition, I would extend a bit the caption that, in the current form, is poorly informative.
Response: We have cited the paper when mentioning prior work on cooperation. We also have improved the figure with labels and arrows as the referee suggested and we also have revised the caption to make it more precise and more informative (see p.6).
- It seems that sign “=“ is missing in eq.(6) and (8)
- In eq. (9) index of summation are missing in the inner part of the equation (I.e. the involved probabilities)
- Please, check eq.(22)
- The quality of figure (2) could be improved. In addition, now \gamma is defined as Selection Strength. Why?
- Please, check lines 139 and 140, maybe a part of the sentence has been removed during the generation of the file.
- Figure 4, the name of axis cannot be read. The resolution should be higher.
Response: We have carefully checked these issues as pointed out by the referee and have made changes and corrections accordingly. We also have improved the quality of all our figures. We thank the referee for a careful reading of our paper (see p.5 – 18, lines 146 - 347).
- In final discussion, I would describe the relation between \alpha and \sigma in relation to the results obtained in the investigation. These two parameters are relevant for the proposed model, so their relation could be interesting.
Response: Thanks for this comment. We have related our theoretical results to our key model parameters, including $\alpha$ and $\sigma$, in the revised discussion part. See the revised manuscript (p. 19-20, lines 349-376).
- It would be interesting a brief discussion/mention about the size of the population, with related results in this model, and the outcomes described in “Evolutionary dynamics of group formation, PLoS ONE 12(11) 2017” where population size has a strong relation with the evolutionary dynamics of a strategy
Response: In response to this comment, we now have mentioned the effect of
population size on cooperation, when we first derive the condition for cooperation, and also cited this paper as pointed out the referee. See the revised manuscript (p. 12, lines 234 – 238).
- Conclusion should be extended, including a wider discussion about the motivations, brief mention about the general model setup and approach to investigate it, and further discussion about points of contact with other fields and related open problems (e.g. biology, ecology, social systems, etc). In principle, Discussion and Conclusion could be merged in only one final section
Response: We have merged discussion and conclusion together in the end of our main text. We have thoroughly rewritten the Discussion part in our revised manuscript. We are grateful to the referee for thoughtful suggestions and helpful comments which have helped us greatly improve our manuscript (see p.19-20, lines 348 - 406).
